# Dynamic representation of partially occluded objects in primate prefrontal and visual cortex

Amber M Fyall[1†], Yasmine El-Shamayleh[2†], Hannah Choi[3], Eric Shea-Brown[4], Anitha Pasupathy[5]*

[1]Department of BIological Structure, Washington National Primate Research Center, University of Washington, Seattle, United States; [2]Physiology and Biophysics, Washington National Primate Research Center, University of Washington, Seattle, United States; [3]Applied Mathematics, University of Washington Institute for Neuroengineering, University of Washington, Seattle, United States; [4]Department of Applied Mathematics, University of Washington, Seattle, United States; [5]Department of BIological Structure, University of Washington, Seattle, United States

**Abstract** Successful recognition of partially occluded objects is presumed to involve dynamic interactions between brain areas responsible for vision and cognition, but neurophysiological evidence for the involvement of feedback signals is lacking. Here, we demonstrate that neurons in the ventrolateral prefrontal cortex (vlPFC) of monkeys performing a shape discrimination task respond more strongly to occluded than unoccluded stimuli. In contrast, neurons in visual area V4 respond more strongly to unoccluded stimuli. Analyses of V4 response dynamics reveal that many neurons exhibit two transient response peaks, the second of which emerges after vlPFC response onset and displays stronger selectivity for occluded shapes. We replicate these findings using a model of V4/vlPFC interactions in which occlusion-sensitive vlPFC neurons feed back to shape-selective V4 neurons, thereby enhancing V4 responses and selectivity to occluded shapes. These results reveal how signals from frontal and visual cortex could interact to facilitate object recognition under occlusion.
DOI: https://doi.org/10.7554/eLife.25784.001

*For correspondence: pasupat@uw.edu

[†]These authors contributed equally to this work

**Competing interests:** The authors declare that no competing interests exist.

## Introduction

When an object is partially occluded, relevant sensory evidence available to the visual system is diminished, making the process of object recognition challenging. Nevertheless, primates are remarkably adept at recognizing partially occluded objects –a common occurrence in the natural world. The neural mechanisms that mediate this perceptual capacity are largely unknown and are the focus of this study.

Biologically inspired models of object recognition are often implemented as hierarchical, feedforward architectures (*Perrett and Oram, 1993*; *Wallis and Rolls, 1997*; *Riesenhuber and Poggio, 1999*) despite extensive evidence for the role of feedback signaling in visual processing (*Lamme et al., 1998*; *Gilbert and Li, 2013*). These feedforward models, and even more elaborate schemes such as artificial convolutional neural networks, remain incapable of successfully recognizing partially occluded objects, although they can perform other object recognition tasks well (*Wyatte et al., 2012*; *Pepik et al., 2015*). The failure of these models has been attributed to the exclusion of critical computations mediated by feedback signals (*Yuille and Kersten, 2006*; *Kriege-skorte, 2015*; *Rust and Stocker, 2010*; *Tang and Kreiman, 2017*). Indeed, more recent models

that incorporate feedback signals show improved recognition performance for occluded objects (*O'Reilly et al., 2013*; *Tang et al., 2014b*). However, little is known about where the relevant feedback signals originate in higher cortex, where they terminate in visual cortex and how they contribute to the recognition of occluded objects. To provide new insights, we investigated the role of the prefrontal cortex in the representation of occluded objects, focusing on how responses in this area compare to responses in the visual cortex, and how frontal and visual cortical areas might interact to facilitate recognition performance.

The prefrontal cortex (PFC) plays an important role in cognitive control—the orchestration of thought and action in accordance with internal goals (*Miller and Cohen, 2001*). Given its high-level function, it may seem unlikely that PFC would contribute to low-level visual representations and mediate the perception and recognition of occluded objects. However, anatomical studies demonstrate that a sub-region of PFC, the ventrolateral PFC (vlPFC), receives direct projections from visual cortical areas involved in higher form processing, that is V4 and inferotemporal cortex (IT) (*Barbas and Mesulam, 1985*; *Ungerleider et al., 2008*). The vlPFC also sends projections back to these visual areas (*Ninomiya et al., 2012*). The existence of functional interactions between these areas is also supported by the demonstration of synchronous neural activity in the theta frequency range between lateral PFC and V4 during perceptual discrimination of visual stimuli (*Liebe et al., 2012*) and by the engagement of PFC in perceptual processing under conditions of greater task difficulty (*Jiang and Kanwisher, 2003*). Given the anatomical and physiological evidence for interactions between vlPFC and visual cortical areas, we hypothesized that vlPFC responses could contribute to the representation and recognition of objects when perceptual judgments are made more difficult by partial occlusion.

To test this hypothesis, we conducted neurophysiological recordings in rhesus monkeys while they discriminated partially occluded shapes. Based on these neuronal data, we addressed three questions. First, how do vlPFC neurons respond to partially occluded shapes compared to neurons in visual area V4? Second, are the response dynamics and tuning properties of V4 neurons consistent with the arrival of feedback signals from vlPFC? Third, is V4 neuronal discriminability for occluded shapes enhanced after the putative arrival of feedback signals from vlPFC?

## Results

### Responses to partially occluded shapes in ventrolateral prefrontal cortex

To determine how vlPFC contributes to the representation and recognition of partially occluded objects, we studied single neuronal responses in monkeys performing a sequential shape discrimination task. In this task, monkeys reported whether two sequentially presented shapes, the 'reference' and 'test', were the same or different by making a saccade to one of two choice targets (*Figure 1A*). To test discrimination under occlusion, the test stimulus was partially occluded with a field of randomly positioned dots. The level of occlusion was titrated by varying dot diameter and was quantified as the percentage of the shape area that remained visible (% visible area). In each session, two shapes were chosen from a standard stimulus set (*Pasupathy and Connor, 2001*; *Kosai et al., 2014*) to serve as the discriminanda. For both monkeys, task performance was high for unoccluded stimuli (100% visible area) and decreased gradually as the % visible area decreased (*Figure 1B*) — that is as occlusion increased (gray arrow).

We analyzed the responses of vlPFC neurons during the test stimulus epoch in which occlusion level was varied. Many neurons responded strongly to occluded stimuli and weakly to unoccluded stimuli. Data from two example neurons are shown (*Figure 2*). The responses of the first example neuron demonstrate a preference for one of the two shapes used (compare *Figure 2A–B*). For both the preferred and non-preferred shape, responses were stronger when the shapes were occluded (colored lines) than unoccluded (black lines). These responses were also more discriminable when the shapes were occluded: shape selectivity was stronger for occluded than unoccluded stimuli (*Figure 2C*; see Materials and methods). The responses of the second example neuron were also stronger when the shapes were occluded than unoccluded (*Figure 2D–F*). However, this neuron showed no preference for either of the two shapes used; shape selectivity was therefore weak for occluded and unoccluded stimuli (*Figure 2F*). The responses of this second example neuron are

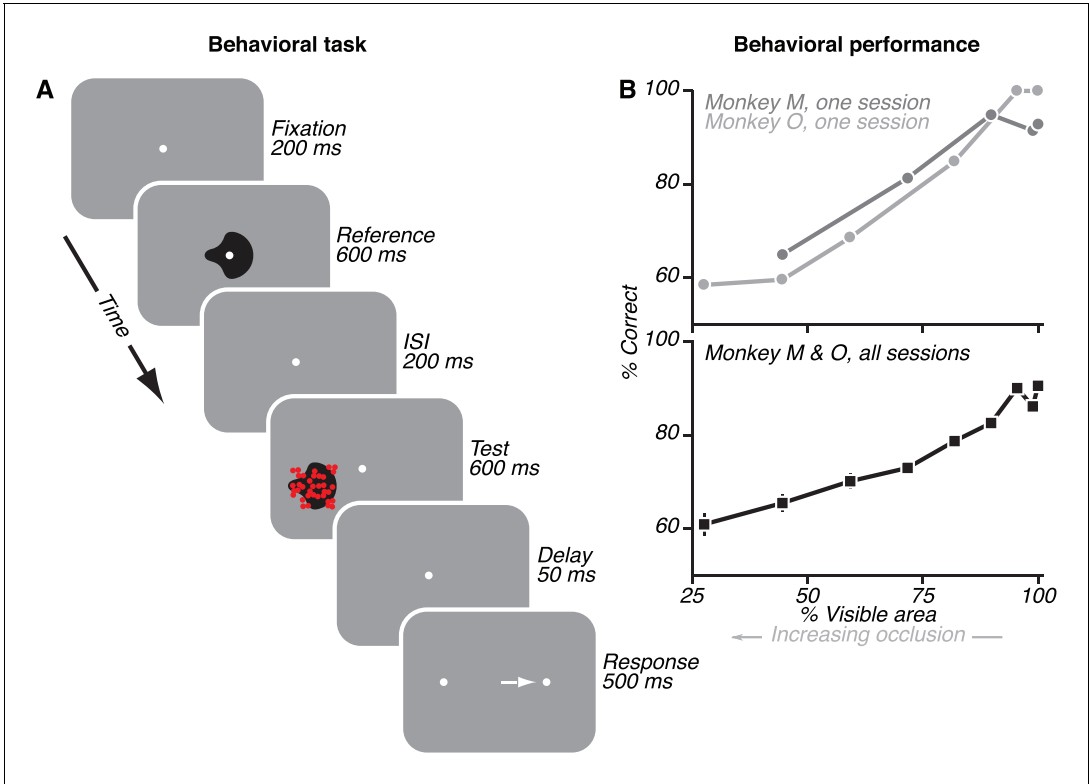

**Figure 1.** Behavioral task and monkey performance. (**A**) On each trial, monkeys viewed two sequentially presented stimuli: an unoccluded shape (reference) and either an unoccluded or a partially occluded shape (test). Monkeys reported whether the two shapes were the same or different by making a saccade to the right or left choice target, respectively. (**B**) Monkey performance on example individual sessions (top) and average performance across all sessions (bottom). Performance was near 100% for unoccluded stimuli (100% visible area) and declined gradually for higher occlusion levels (*i.e.* <100% visible area).

DOI: https://doi.org/10.7554/eLife.25784.002

consistent with sensitivity for the total area or circumference of the occluding dots. In contrast, the responses of the first example neuron are inconsistent with this interpretation because the stronger responses to occluded stimuli were also accompanied by stronger shape selectivity.

Most of the vlPFC neurons we recorded responded more strongly to occluded stimuli (*Figure 3*). Of 216 neurons that were visually responsive during the test stimulus epoch (see Materials and methods), 98 neurons (45%) were significantly modulated by occlusion level (2-way ANOVA, p<0.05). The responses of most of these occlusion-sensitive neurons were stronger for higher occlusion levels (*Figure 3A*). For individual neurons, this observation manifested as a negative linear regression slope between % visible area and the average responses during the test epoch (*Figure 3B*). Of the 98 occlusion-sensitive neurons, 71 had a negative regression slope; 59 neurons had a slope that was significantly less than zero (p<0.05). For the subset of occlusion-sensitive neurons, normalized responses were also stronger at higher occlusion levels (*Figure 3C*). The results were also qualitatively similar when all visually responsive vlPFC neurons were included.

Shape selectivity was stronger for occluded than unoccluded stimuli across the population of vlPFC neurons. For the subset of occlusion-sensitive vlPFC neurons, shape selectivity was strongest for occluded stimuli at intermediate occlusion levels (blue/green) and weakest for unoccluded stimuli (black) (*Figure 3D*). This observation also held for the subset of shape-selective vlPFC neurons (N = 66; *Figure 3—figure supplement 1B*) and for all visually responsive neurons (N = 216; *Figure 3—figure supplement 1D*). Even for the small subset of vlPFC neurons that responded more strongly or equally well to unoccluded stimuli (27/98 neurons had a positive regression slope; 17/98 neurons had a slope significantly greater than zero, p<0.05), shape selectivity was not stronger for

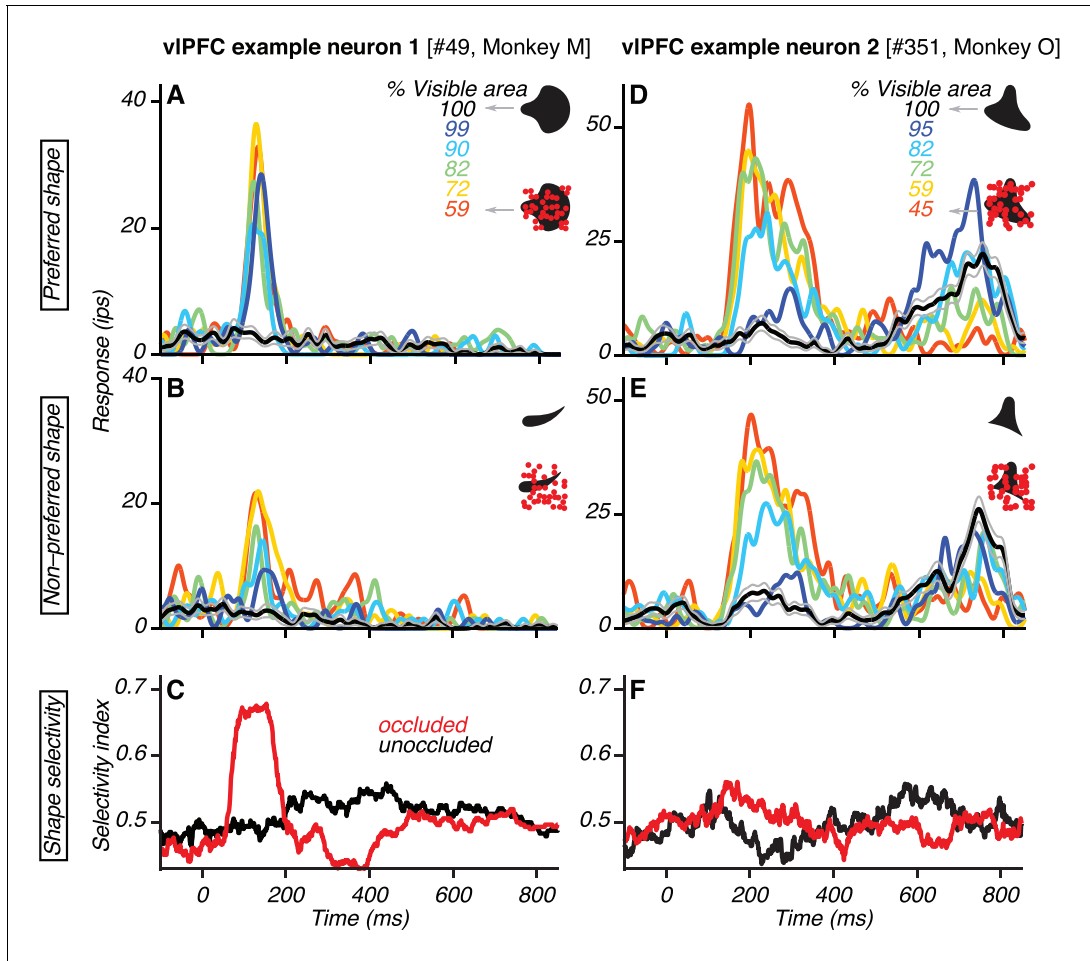

**Figure 2.** Responses of example vlPFC neurons. (A–B) Responses of one neuron to its preferred and non-preferred shape (compare A and B) at different occlusion levels (colors). Responses to unoccluded stimuli were weak (black; gray lines show standard error of the mean) whereas responses to occluded stimuli were stronger (colors). For PSTHs, σ was 10 ms. (C) Neuronal shape selectivity across time for unoccluded (black) and occluded (red) stimuli quantified by a sliding window ROC curve analysis (see Materials and methods). (D–E) Responses of a second vlPFC neuron, showing comparable responses to both shapes (compare D and E). Responses to unoccluded stimuli were weak whereas responses to occluded stimuli were stronger. The test stimulus was extinguished at 600 ms and the elevation in response beyond this time point was due to the saccadic response. (F) Neuronal shape selectivity across time (same format as in C).

DOI: https://doi.org/10.7554/eLife.25784.003

unoccluded than occluded stimuli (see *Figure 3—figure supplement 2A*). Thus, the vlPFC neuronal population has stronger, more shape-selective responses to occluded than unoccluded stimuli.

In addition to showing a preference for occluded stimuli during the test epoch, the responses of many vlPFC neurons during this epoch also signaled whether the test and reference shapes were a match/nonmatch. Of 216 neurons, the responses of 42 had a significant main effect of match/nonmatch condition, and the responses of 65 other neurons had a significant interaction between shape and match/nonmatch condition (two-way ANOVA, $p < 0.05$).

The vlPFC results presented thus far differ markedly from what we and others have reported in the visual cortex regarding the representation of occluded and unoccluded objects. In monkey cortical areas V4 (*Kosai et al., 2014*) and IT (*Kovács et al., 1995*) and in human occipitotemporal cortex (*Tang et al., 2014a*), neuronal responses are strongest for unoccluded objects and neuronal shape selectivity declines gradually with increasing occlusion level. Thus, the strong responses and shape selectivity of vlPFC neurons for occluded stimuli cannot be inherited directly from visual cortex. Next, we examine how the signals in vlPFC compare to those in the visual cortex by analyzing

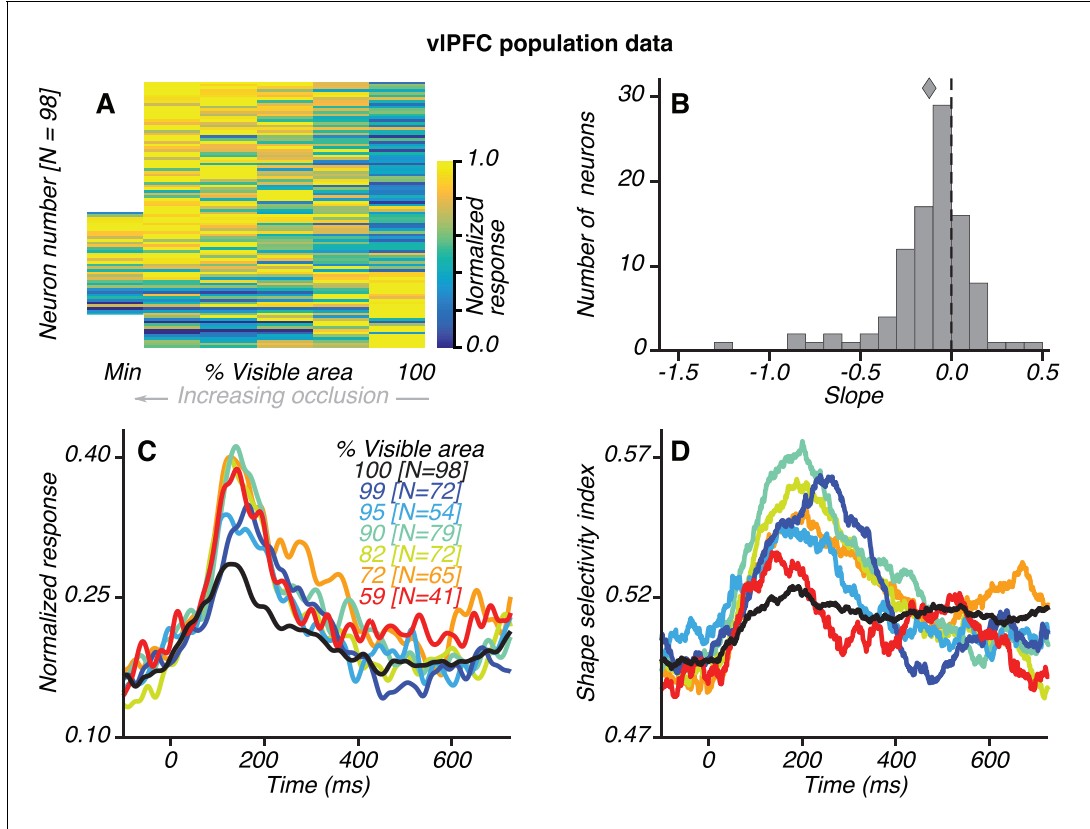

**Figure 3.** Population results for vlPFC neurons. (A) Normalized average responses (80–230 ms) to preferred shapes across occlusion level (columns). Along the abscissa, % visible area is rank ordered; so each column could represent a different occlusion level for each neuron. From left to right, occlusion level decreases and % visible area increases, with 100% representing the unoccluded case. Neurons were tested with five or six occlusion levels (62/98 and 36/98 neurons, respectively), including the unoccluded case. Responses were normalized by the maximum response for each neuron, separately. For most neurons, responses were weak (blue) for unoccluded stimuli and stronger (yellow) for higher occlusion levels. Responses to the unoccluded stimuli were strongest for only 17/98 neurons. (B) Distribution of linear regression slopes fit to the responses in A. Most neurons (71/98) had negative slopes, indicating stronger responses to occluded stimuli; the median slope was −0.12. (C) Normalized average responses derived from the responses of occlusion-sensitive neurons to the preferred shapes at different occlusion levels. vlPFC neurons were studied at 5–6 oc clusion levels chosen from a set of 9 possible values (100, 99, 95, 90, 82, 72, 59, 45% and 27% visible area). The occlusion levels presented to each neuron was different, and not all neurons were studied at all the occlusion levels listed. The numbers of neurons contributing to each curve are listed in brackets (see Materials and methods). Data for the highest occlusion levels tested (45% and 27% visible area) are not shown because too few neurons contributed to the population average and the curves for these occlusion levels were highly variable. Across the population, responses were weakest to unoccluded stimuli (black) and stronger for intermediate and high occlusion levels. (D) Average shape selectivity as a function of occlusion level. Shape selectivity was weak for unoccluded stimuli (black) at all time points whereas it gradually increased for occluded shapes (colors), reaching maximal values between ~100–300 ms. Also see *Figure 3—figure supplement 1* and *Figure 3—figure supplement 2*.

DOI: https://doi.org/10.7554/eLife.25784.004

The following figure supplements are available for figure 3:

**Figure supplement 1.** Shape selectivity as a function of occlusion for different groups of vlPFC neurons.

DOI: https://doi.org/10.7554/eLife.25784.005

**Figure supplement 2.** Shape selectivity for two groups of vlPFC neurons.

DOI: https://doi.org/10.7554/eLife.25784.006

neuronal response dynamics in V4 datasets collected previously (*Kosai et al., 2014*), recorded in the same monkeys while they performed the same behavioral task used in the vlPFC testing sessions.

## Responses to partially occluded shapes in visual area V4

If feedback signals originating in vlPFC contribute to V4 responses, their influence on V4 would be evident after the onset of vlPFC responses. Additionally, their influence would manifest in V4 as stronger responses for occluded than unoccluded stimuli, consistent with the vlPFC response

properties described earlier. Many V4 neurons in our dataset did *not* show evidence of feedback modulation in their temporal response profiles. The responses of one such example neuron (*Figure 4A–B*) had a temporal response profile with a single transient response phase (i.e. peak) followed by a sustained response phase. During both the transient and sustained phases, responses were stronger for unoccluded than occluded stimuli, unlike what we observed in vlPFC. However, many other V4 neurons showed a different temporal profile with two transient response peaks – one early and one late – each of which showed a different dependency on occlusion level. The responses of one such example neuron (*Figure 4C–D*) had two transient response peaks: the first ~82 ms and the second ~150 ms after test stimulus onset. The neuron's responses during the first peak (*Figure 4D*, black bar) were strongest for the unoccluded stimulus and declined gradually with increasing occlusion level. In contrast, the neuron's responses during the second peak (*Figure 4D*, red bar) were strongest for intermediate occlusion levels.

The responses of two other V4 neurons with two peaks are shown (*Figure 5*). For one neuron (*Figure 5A*), the first and second peaks occurred ~63 ms and ~191 ms after stimulus onset. For the other neuron (*Figure 5B*), the first and second peak occurred ~66 ms and ~218 ms after stimulus onset. Additional examples of V4 neurons with two peaks are provided (*Figure 5—figure supplement 1*).

We developed an *ad hoc* peak finding algorithm to identify V4 neurons with and without two transient response peaks (see Materials and methods; *Figure 6—figure supplement 1*). The algorithm detected the occurrence of two robust transient peaks separated by a sizeable intervening trough, and the results were vetted using statistical tests. Of 85 neurons, 30 neurons (35%; 14 neurons recorded in Monkey O and 16 neurons recorded in Monkey M) were classified as having two peaks (*Figure 6A*) and 55 neurons were classified as not having two peaks (*Figure 6B*). The second response peak was less striking when the responses were averaged across neurons (*Figure 6C*) due to variability in second peak times for individual neurons (see also *Figure 10—figure supplement 6*). Across all V4 neurons with two peaks, the timing of the first and second peaks had a broad range, with a median of 84 ms and 214 ms, respectively (*Figure 6C*). In comparison, across all occlusion-sensitive vlPFC neurons, the peak response occurred later than in V4, 93–581 ms after test

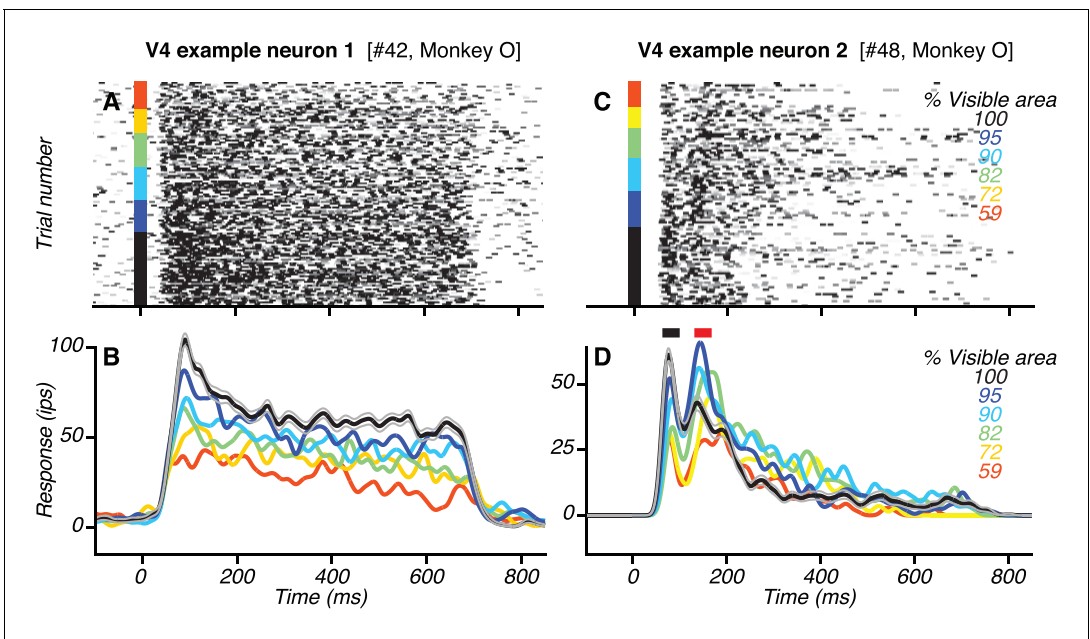

**Figure 4.** Responses of example V4 neurons. (A–B) Responses (rasters and PSTHs) of one neuron to the preferred shape at different occlusion levels (colors). This neuron had only one transient response peak. Responses were strongest to the unoccluded stimulus (black; gray lines show standard error of the mean) and declined gradually with increasing occlusion level (i.e. lower % visible area). (C–D) Responses of another V4 neuron. This neuron had two transient response peaks (black and red horizontal bars in (D) with different dependence on occlusion level.
DOI: https://doi.org/10.7554/eLife.25784.007

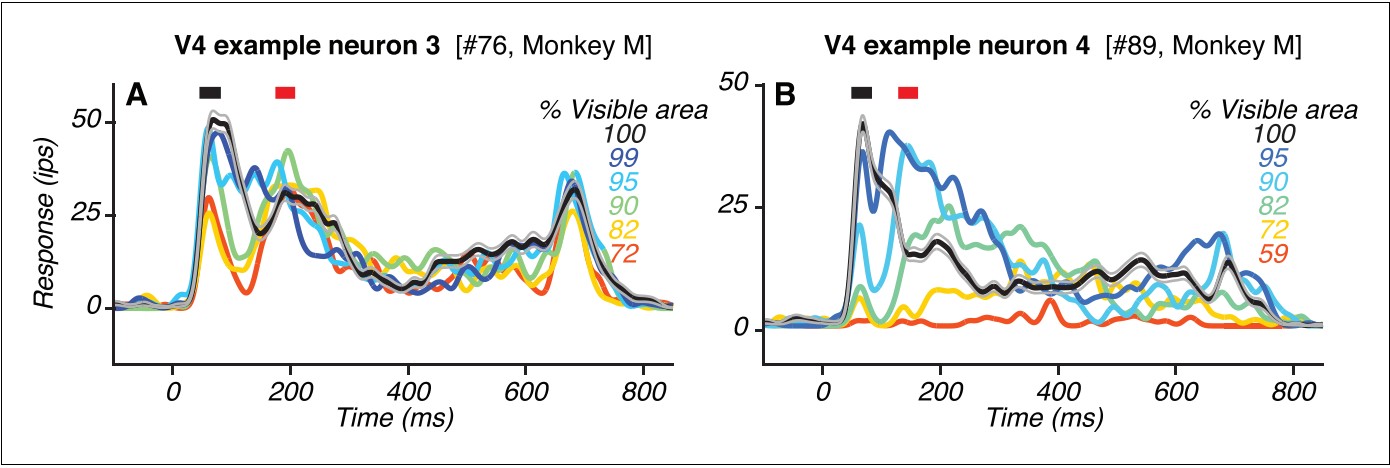

**Figure 5.** Responses of two additional example V4 neurons with two transient response peaks. (A–B) PSTHs of responses to the preferred shape at different occlusion levels (colors; same format as in *Figure 4B, D*). Also see *Figure 5—figure supplement 1*.
DOI: https://doi.org/10.7554/eLife.25784.008
The following figure supplement is available for figure 5:

**Figure supplement 1.** Additional examples of V4 neurons with two peaks.
DOI: https://doi.org/10.7554/eLife.25784.009

stimulus onset, with a median of 157 ms (*Figure 6D*). Thus, the median peak time in vlPFC straddled the median peak times of the first and second response peaks in V4.

If feedback signals from vlPFC contribute to V4 activity during the second response peak, we expect that V4 responses would differ in their dependence on occlusion level over time. We therefore assessed neuronal sensitivity to occlusion during the first and second response peaks in V4. Data from three example V4 neurons are shown (*Figure 7A–C*). For the first neuron (*Figure 7A*; same neuron as in *Figure 4C–D*), responses during the first peak (black) declined gradually as occlusion level increased. In contrast, responses during the second peak were strongest at intermediate occlusion levels. Thus, the difference in responses between the first and second peak (gray) was largest at intermediate occlusion levels. The two other example neurons showed similar results (*Figure 7B–C*; same neurons as in *Figure 5A–B*, respectively): the difference in responses between the first and second peak was larger for occluded stimuli than unoccluded stimuli for both neurons.

We compared the first and second peak responses for V4 neurons with and without two peaks. For both groups of neurons, responses during the first peak (69–99 ms) declined gradually as occlusion level increased. There was no significant difference between neurons with and without two peaks in their responses during the first peak at any occlusion level (t Test, p>0.1). However, later in the test stimulus epoch (199–229 ms), around the time of the second peak, the responses of the two groups of neurons had different trends. For neurons with two peaks, the difference in responses between the first and second peak was largest for intermediate occlusion levels and small for unoccluded stimuli and high occlusion levels (*Figure 7D*, dark gray). Thus, the response difference curve had an inverted U shape, as seen for the example neurons (compare dark gray curves, *Figure 7A–D*). In contrast, for neurons without two peaks, the difference in responses between the two time points was small and similar in magnitude across all occlusion levels (*Figure 7D*, light gray). The difference in responses between the first and second peaks was significantly greater for neurons with two peaks than other neurons at intermediate occlusion levels (compare curves in 7D; t Test, p<0.05, asterisks). This finding is explained by the observation that V4 neurons without two peaks had responses that declined gradually over time, for all occlusion levels (*Figure 6B*). In contrast, neurons with two peaks showed a relative increase in responses to occluded stimuli during the second peak (*Figure 6A*), a pattern that mirrored the responses of occlusion-sensitive neurons in vlPFC.

To quantify how shape selectivity evolves during the test stimulus epoch, we examined average neuronal shape selectivity across time for unoccluded and occluded stimuli for V4 neurons with two peaks (*Figure 8A*) and those without (*Figure 8B*). For unoccluded stimuli (black lines), shape

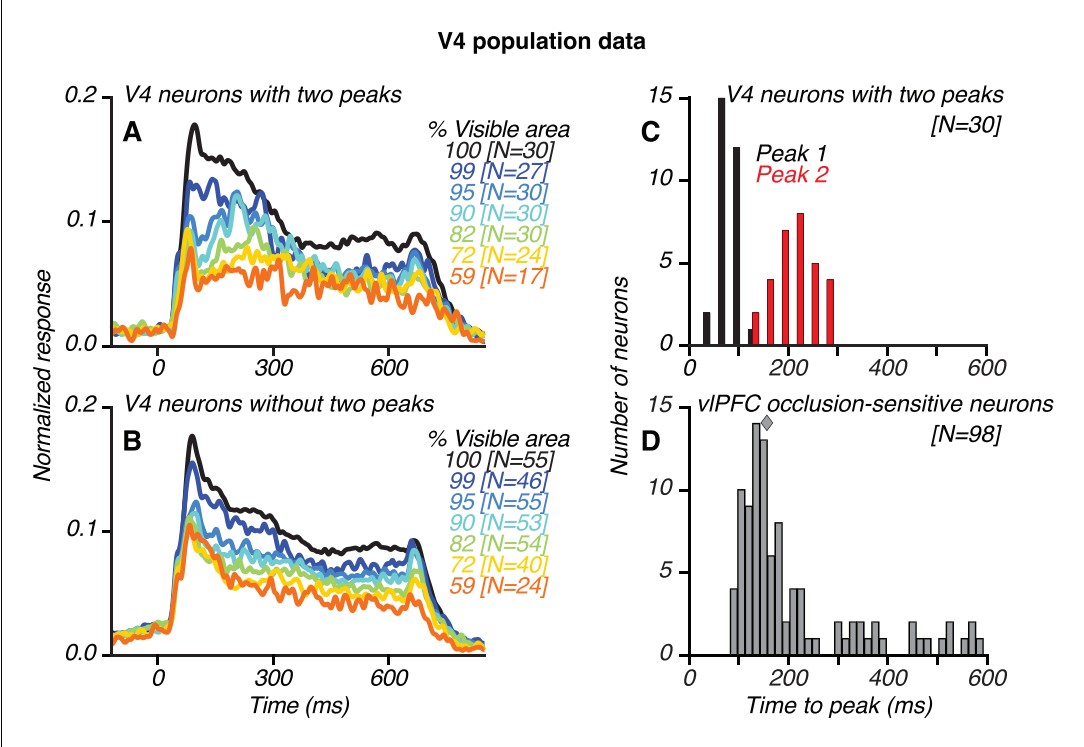

**Figure 6.** Population results for V4 and vlPFC neurons. (A–B) Population-level, normalized PSTHs for 30/85 V4 neurons that showed two transient response peaks (A) and for V4 neurons without two peaks (B), identified using an *ad hoc* algorithm (see Materials and methods). The numbers of neurons contributing to each curve are listed in brackets. (C) Distribution of the first (black) and second (red) response peak times for V4 neurons with two peaks. (D) Distribution of the latency of peak responses for occlusion-sensitive vlPFC neurons; median latency was 157 ms (diamond). The first and second response peaks in V4 typically occurred before and after the peak response of vlPFC neurons, respectively. Also see *Figure 6—figure supplement 1*.

DOI: https://doi.org/10.7554/eLife.25784.010

The following figure supplement is available for figure 6:

**Figure supplement 1.** Schematic of the *ad hoc* peak finding algorithm used.

DOI: https://doi.org/10.7554/eLife.25784.011

selectivity was similar in magnitude and time course for the two groups, reaching a maximum value at ~120 ms. For occluded stimuli (colored lines), shape selectivity was similar for the two groups (t Test, p=0.6) early in the test stimulus epoch, around the time of the first peak (69–99 ms). However, shape selectivity for neurons with two peaks was significantly stronger (t Test, p<0.01) later in the test stimulus epoch, around the time of the second peak (199–229 ms). This is because for neurons with two peaks, shape selectivity for occluded stimuli increased over time and reached a maximal value closer to the time of the second peak (*Figure 8—figure supplement 1*).

We demonstrate this enhanced shape selectivity for occluded stimuli later in the test stimulus epoch in two ways. First, we compared the magnitude of shape selectivity at different periods (*Figure 8C*, early and late). Second, we compared the timing of peak selectivity for occluded and unoccluded stimuli (*Figure 8D–E*). For neurons with two peaks, shape selectivity around the time of the second peak was significantly stronger than around the time of the first peak (t Test, p<0.01; *Figure 8C*). This observation did not hold for neurons without two peaks (p=0.92) or for unoccluded stimuli for either group of neurons (p>0.5). The timing of maximal shape selectivity for unoccluded stimuli occurred significantly earlier than the second peak (*Figure 8D*, median 131 vs. 214 ms, respectively; t Test, p<0.01). In contrast, the timing of maximal shape selectivity for occluded stimuli occurred around the time of the second peak (*Figure 8E*, median 188 vs. 214 ms, respectively; t Test, p>0.98),

The enhanced shape selectivity for occluded stimuli around the time of the second peak occurred even for neurons that had stronger responses during the first peak than the second peak (*Figure 8—*

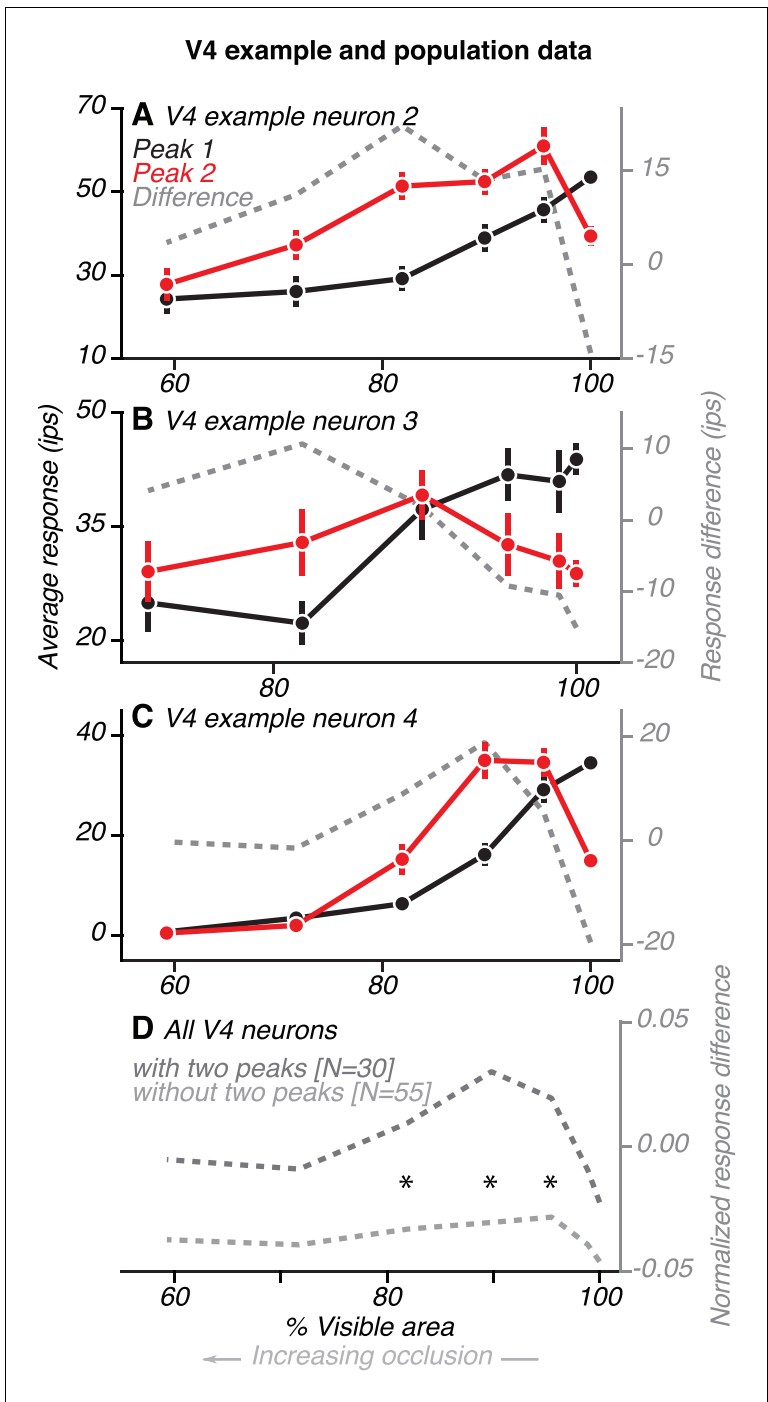

**Figure 7.** Comparison of V4 responses during the first and second peaks. (**A**) Data from one example V4 neuron with two peaks (same neuron as in *Figure 4D*). Average responses (left ordinate) were measured within a 30 ms window centered on the first and second peaks, relative to the preceding baseline (85–115 ms before test stimulus onset). Difference in responses (right ordinate) between the first and second peaks (gray line) is shown. During the first peak (black), responses were strongest for the unoccluded shape (100% visible area) and declined for higher occlusion levels (<100% visible area). During the second peak (red), responses were strongest for intermediate levels of occlusion. Response difference was an inverted U-shaped curve, with a negative value for the unoccluded stimulus and positive for all levels of occlusion. (**B–C**) Data for two other V4 neurons (same neurons as in *Figure 5A–B*, respectively). (**D**) Population average of the difference in responses between first and second peak for neurons with (dark gray) and without (light gray) two peaks. Responses of each neuron were first normalized by the maximum across time and occlusion levels. Then, for each neuron and at each occlusion level, we measured

*Figure 7 continued on next page*

*Figure 7 continued*

the difference in response between two time points: during the initial transient (69–99 ms after test stimulus onset) and at a later time in the sustained response phase (199–229 ms after test stimulus onset). These time points are centered around the median peak time of the first and second peaks, respectively. For neurons with two peaks, the average response difference between first and second peaks followed a U-shaped curve, with positive values for intermediate levels of occlusion. For neurons without two peaks, the difference curve was flat and negative for all levels of occlusion. Asterisks mark occlusion levels for which the average response difference curve was significantly greater for neurons with two peaks (t Test, $p<0.05$).
DOI: https://doi.org/10.7554/eLife.25784.012

*figure supplement 2A*). This finding suggests that response magnitude during the second peak does not fully account for the strength of shape selectivity. However, for neurons that had stronger shape selectivity during the second peak, the relative magnitude of responses during the second peak was larger for the preferred than non-preferred shapes (*Figure 8—figure supplement 2B*). This differential enhancement of responses to preferred shapes serves to amplify shape selectivity during the second peak.

Given that we classified neurons based on an *ad hoc* algorithm with customized parameters (see Materials and methods and *Figure 6—figure supplement 1*), we sought to ensure that the findings did not depend on the choice of parameters used and that the algorithm did not yield false-positives. To address these concerns, we examined population results for neurons with and without two peaks using different choices of threshold parameters (*Figure 8—figure supplements 3–4*). Additionally, we developed a model-based procedure that was independent of the *ad hoc* peak finding algorithm to identify neurons whose responses to occluded stimuli were stronger than expected from a linear scaling of responses to unoccluded stimuli (*Figure 8—figure supplements 5–6*). We found good correspondence between the model-based and algorithm-based approaches in terms of the neurons identified as having two peaks. Population results generated using different parameter choices for the *ad hoc* algorithm and using the model-based procedure were remarkably similar to those presented earlier (*Figures 6* and *8*).

Collectively, these results support the hypothesis that occlusion-sensitive signals in vlPFC are relayed to V4 and that these feedback signals contribute to V4 responses during the second peak, enhancing neuronal selectivity for occluded shapes. These putative feedback signals may be well suited to enhance perceptual discriminability of partially occluded objects.

## A model of V4–vlPFC interactions

To demonstrate the plausibility of feedback signals from vlPFC contributing to V4 responses to occluded stimuli, we constructed a two-layer dynamical model of V4 and vlPFC interactions (*Figure 9*; see Materials and methods). In this model, shape-selective V4 units send feedforward inputs to vlPFC units (*Figure 9*, light gray arrows). The shape preference of each vlPFC unit is inherited from the V4 unit which provides the strongest input. vlPFC units also receive a gain modulation signal that increases with increasing occlusion level (dashed box), imparting a preference for occluded stimuli that is not observed in the feedforward V4 inputs to vlPFC. Additionally, vlPFC units send feedback inputs onto V4 units (medium gray arrows) with connection strengths that are proportional to the feedforward signals from each V4 unit. Importantly, feedback signals from vlPFC first pass through a rectifying nonlinearity prior to their arrival in V4 (*Equation 7*, Materials and methods). The vlPFC feedback signals contribute to two key response features of the V4 units: a second transient response peak and a dynamic preference for occlusion level over the test stimulus epoch.

To demonstrate model performance, we present data for a simulated V4 and vlPFC unit (*Figure 10*). In the model, feedforward input to the V4 unit is modulated both by shape and occlusion level (*Figure 10A*): it is strongest when the preferred shape is unoccluded and is progressively weaker for higher occlusion levels. This pattern is consistent with our V4 neuronal data and captures responses during the first peak for V4 neurons with two peaks, as well as the responses of V4 neurons without two peaks. Occlusion-dependent gain modulation of feedforward input from V4 produced vlPFC unit responses that were weak to unoccluded stimuli and stronger to occluded stimuli (*Figure 10B*). Furthermore, the V4 unit receiving feedback signals from vlPFC had two transient response peaks: one earlier and one later than the response peak of the vlPFC unit (*Figure 10C*).

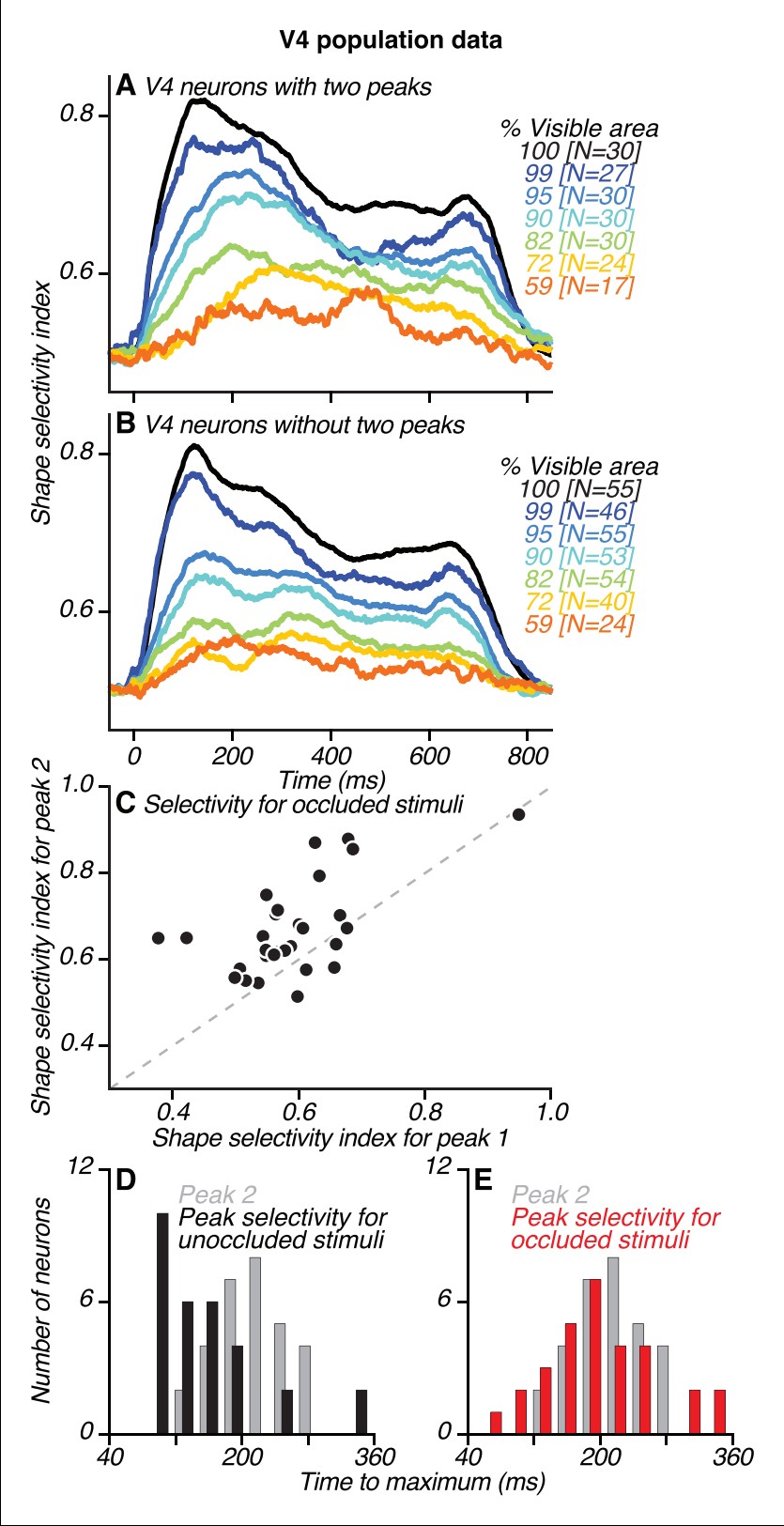

**Figure 8.** Dynamics of neuronal shape selectivity in V4. (A–B) Time course of average shape selectivity for unoccluded (black) and occluded (colors) stimuli for V4 neurons with two peaks (A) and V4 neurons without two peaks (B). For unoccluded stimuli (black), the time course and magnitude of shape selectivity were similar for the

*Figure 8 continued*

two neuronal groups. For stimuli at intermediate occlusion levels (72–95% visible area), shape selectivity was stronger and reached a maximal value later for neurons with two peaks (compare corresponding colors in **A** and **B**, also see *Figure 8—figure supplement 1*). (**C**) For most neurons with two peaks, shape selectivity at the time of the second peak (ordinate) was stronger than during the first peak (abscissa). Shape selectivity for occluded stimuli was significantly higher during the second peak than during the first peak (t Test, p<0.01). We computed shape selectivity using ROC analysis across all occlusion trials (i.e. % visible area <100) based on spike counts in a 30 ms window centered around the times of the first and second peaks, respectively (also see *Figure 8—figure supplement 2*). (**D–E**) For neurons with two peaks, the time of the second peak (gray) is compared to the time of maximal shape selectivity (black and red), separately for unoccluded (black, **D**) and occluded stimuli (red, **E**). Shape selectivity for unoccluded stimuli reached a maximal value significantly earlier than the second peak (t Test, p<0.01); this was not the case for occluded stimuli (t Test, p=0.98). Also see *Figure 8—figure supplements 3–6*.

DOI: https://doi.org/10.7554/eLife.25784.013

The following figure supplements are available for figure 8:

**Figure supplement 1.** Comparison of shape selectivity under occlusion for V4 neurons with and without two peaks.
DOI: https://doi.org/10.7554/eLife.25784.014

**Figure supplement 2.** Relative response amplitude versus change in shape selectivity during the second peak for V4 neurons with two peaks.
DOI: https://doi.org/10.7554/eLife.25784.015

**Figure supplement 3.** Distribution of parameter values for the peak-finding algorithm.
DOI: https://doi.org/10.7554/eLife.25784.016

**Figure supplement 4.** Dependence of observed V4 results on the peak-finding algorithm parameters.
DOI: https://doi.org/10.7554/eLife.25784.017

**Figure supplement 5.** Analysis of V4 response dynamics across occlusion level.
DOI: https://doi.org/10.7554/eLife.25784.018

**Figure supplement 6.** Population results for neurons with stronger responses to occluded stimuli than expected from linear scaling (see *Figure 8—figure supplement 5*).
DOI: https://doi.org/10.7554/eLife.25784.019

The V4 unit's responses during the first peak were strongest for the unoccluded stimulus and declined with increasing occlusion level. In contrast, the V4 unit's responses during the second peak were strongest at intermediate occlusion levels. Thus, the response dynamics generated by this V4–vlPFC interaction model successfully recapitulated the dynamics observed in our neuronal recordings (*Figures 2* and *4*).

## Parsimonious model construction

Our model was constructed to include only the minimal set of mechanisms that were needed to account for the main features of the neurophysiological data. We started with the simplest feedforward model composed of two V4 units and two vlPFC units, and we included four mechanisms to achieve the desired dynamics in V4 and vlPFC responses: (1) feedback from vlPFC to V4; (2) synaptic adaptation in the feedforward inputs from V4 to vlPFC; (3) half-wave rectification of feedback signals from vlPFC to V4; (4) occlusion-dependent gain modulation input to vlPFC. We included feedback from vlPFC to V4 because a network with only feedforward connections from V4 to vlPFC units cannot generate the second response peak in V4 units (*Figure 10—figure supplement 1A*). Thus, in our model, feedback from vlPFC to V4 is necessary to reproduce the response dynamics observed in V4. Second, without synaptic adaptation on the feedforward connections from V4 to vlPFC (*Equation 9*, Materials and methods), the feedforward-feedback loop reinforces activity in V4 and PFC units positively (*Figure 10—figure supplement 1B*). The resulting 'ringing' and 'blow up' in simulated model responses are inconsistent with the data, thus arguing for the inclusion of an adaptation mechanism. Indeed, such an adaptation mechanism is often used in models to soften positive feedback loops (e.g. *Wei and Wang, 2016*). Third, without half-wave rectification of the feedback input from vlPFC to V4 units, the model produces a large second response peak even to presentation of non-preferred stimuli (*Figure 10—figure supplement 2*), which conflicts with the data (*Figure 8—figure supplement 2B*). Thus, without half-wave rectification, the enhanced shape selectivity during the second peak in the V4 neuronal data (*Figure 8C*) is not reproduced by the model. Given that

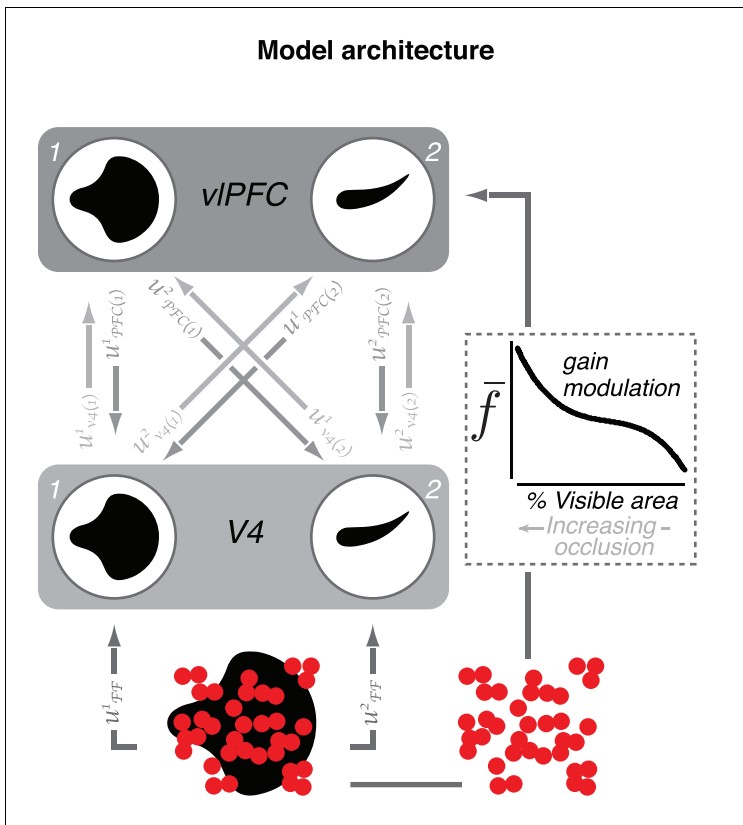

**Figure 9.** Model of V4–vlPFC interactions. The two-layer dynamical network model includes two V4 units (1 and 2) and two vlPFC units (1 and 2). The V4 units prefer two different shapes. Each vlPFC unit receives inputs from both V4 units; the shape preference of a vlPFC unit is determined by the preference of the V4 unit that provides the stronger input. V4 units receive feedforward input from upstream visual areas denoted by $u^i_{FF}$ and feedback from both vlPFC units denoted by $u^i_{PFC(j)}$. In this notation, $i$ indexes the recipient unit and $j$ indexes the sender unit. Feedback inputs are passed through a rectifying nonlinearity prior to arrival in V4. Each vlPFC unit receives feedforward inputs from both V4 units, denoted by ($u^i_{V4(j)}$), as well as a modulatory signal, $\bar{f}$, that depend on occlusion level (for modeling details, see Materials and methods).
DOI: https://doi.org/10.7554/eLife.25784.020

rectifying nonlinearities can occur when synaptic inputs are transformed into output spikes, the model suggests that feedback from vlPFC may arrive in V4 after passing through a synapse. Indeed, anatomical observations of disynaptic feedback connections between V4 and vlPFC exist (*Ninomiya et al., 2012*). Fourth, when all other mechanisms are in place but the gain modulation is removed, vlPFC responses decrease with increasing occlusion level (*Figure 10—figure supplement 1C*). Consequently, V4 shape selectivity under occlusion is not enhanced during the second peak. To consider the possibility that gain modulation of vlPFC may be mediated by signals from V4 or IT cortex, we verified that our simulations were unaffected by delays of up to ~50 ms in the arrival of gain modulation relative to the arrival of shape selective signals (*Figure 10—figure supplement 3*).

## Heterogeneity in model V4 and vlPFC responses

To generate the simulated responses shown (*Figure 10*), we chose model parameters that reproduced the response dynamics of example neurons (*Figures 2A*, *4* and *5*). However, V4 neurons show substantial diversity in the magnitude and timing of the second response peak (*Figure 5—figure supplement 1*). vlPFC neurons also show diversity in terms of their shape selectivity and the dependence of their responses on occlusion level. Therefore, we systematically varied the model parameters governing synaptic strengths and delays to generate diverse simulated response dynamics and patterns in V4 and vlPFC model units. We varied the relative strengths of the feedforward

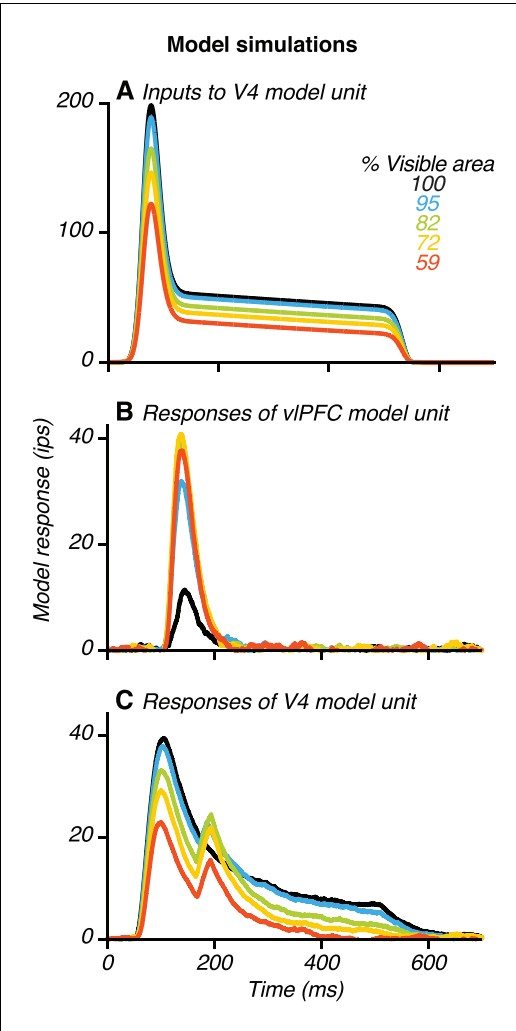

**Model simulations**

**A** *Inputs to V4 model unit*

% Visible area
100
95
82
72
59

**B** *Responses of vlPFC model unit*

Model response (ips)

**C** *Responses of V4 model unit*

Time (ms)

**Figure 10.** Example model results. (**A**) Input to a V4 unit, $u_{FF}^1$, when its preferred stimulus was presented at different occlusion levels (colors). The input was strongest for the unoccluded stimulus (black) and declined gradually with increasing occlusion level. (**B**) Responses of a vlPFC unit. Responses were weak for the unoccluded stimulus and stronger for occluded stimuli. (**C**) Responses of a V4 unit receiving feedback from the vlPFC unit (in **B**) showing two transient peaks. Also see *Figure 10—figure supplements 1–6*.
DOI: https://doi.org/10.7554/eLife.25784.021

The following figure supplements are available for figure 10:

**Figure supplement 1.** Demonstrating the necessity of feedback, synaptic adaptation and gain mechanisms in the model.
DOI: https://doi.org/10.7554/eLife.25784.022

**Figure supplement 2.** Demonstrating the necessity of half-wave rectification in the model.
DOI: https://doi.org/10.7554/eLife.25784.023

**Figure supplement 3.** Effect of delaying the arrival of gain modulation signals to vlPFC in the model.
DOI: https://doi.org/10.7554/eLife.25784.024

input from the two V4 units to vlPFC, and we verified that a second response peak was observed in the V4 unit responses even when vlPFC units are only weakly shape-selective (*Figure 10—figure supplement 4*). By varying the feedback connection strengths, and the synaptic delays between the two areas, we were able to generate a range of second peak magnitudes (*Figure 10—figure supplement 5A*) and second peak times (*Figure 10—figure supplement 5B*). Finally, the population average response across model V4 units (*Figure 10—figure supplement 6*) resembles the V4 population data (*Figure 6A*).

## Discussion

To determine the contributions of prefrontal cortex to the representation and recognition of partially occluded objects, we compared the response dynamics of vlPFC and V4 neurons in monkeys discriminating shapes in the presence and absence of occluders. Our study provides three new insights. First, neuronal responses in vlPFC are strongest for occluded stimuli and weaker for unoccluded stimuli, in contrast to neuronal responses in visual areas V4 and IT (*Kosai et al., 2014*; *Kovács et al., 1995*; *Tang et al., 2014a*). Second, the responses of many V4 neurons have two transient peaks, the second of which emerges after the onset of vlPFC responses and shows a stronger preference for occluded stimuli. Third, neuronal shape selectivity for occluded stimuli in V4 is enhanced during the second transient peak. Our results support the hypothesis that feedback signals from vlPFC mediate V4 responses during the second transient peak and that these signals facilitate object recognition under occlusion.

### Representation of occluded stimuli in vlPFC

Our results demonstrate that visual representations in vlPFC do not always mirror representations in visual cortex and suggest that vlPFC may play an important role in representing objects. We used different experimental approaches for the V4 and vlPFC recordings, but these methodological differences cannot account for differences in how V4 and vlPFC neurons represent occluded and unoccluded stimuli. In V4 recording sessions, but not in vlPFC sessions, we tailored the stimulus color and shape to the preferences of the neuron; this may explain the preponderance of V4 neurons that responded preferentially to unoccluded shapes in our dataset. Without tailoring stimuli we would expect a roughly equal

*Figure 10 continued*

**Figure supplement 4.** Relationship between the strength of shape selectivity in vlPFC in the model and the magnitude of V4 second peak responses produced.

DOI: https://doi.org/10.7554/eLife.25784.025

**Figure supplement 5.** Heterogeneity of V4 second peak response properties reproduced by the model.

DOI: https://doi.org/10.7554/eLife.25784.026

**Figure supplement 6.** Population average of 258 V4 model units with two peaks.

DOI: https://doi.org/10.7554/eLife.25784.027

proportion of neurons showing responses that increased and decreased with increasing occlusion level. We found, however, that 72% of vlPFC neurons responded preferentially to occluded shapes, a proportion that deviates significantly from the null hypothesis (binomial test, p<0.01). Furthermore, because the visible difference between any two shapes declines with increasing occlusion level, we expect shape selectivity to decline regardless of whether we tailored visual stimuli. The enhanced shape selectivity we observed in vlPFC under occlusion defies this expectation.

The stronger responses to occluded stimuli in vlPFC cannot be attributed to neuronal preferences for the color of the occluding dots. We verified in control experiments that the preference for occluded stimuli was independent of dot color (data not shown). Given that many vlPFC neurons are selective for the shape of the occluded stimulus, it is unlikely that vlPFC responses **solely** reflect task difficulty level or attentional demands. If difficulty or attention could fully explain vlPFC responses, occlusion-sensitive neurons would not be shape-selective (i.e. the PSTHs in *Figure 2A* and *Figure 2B* would be identical). We also verified in control experiments that vlPFC neuronal responses were weaker when the occluding dots were in the same color as the background – an observation suggesting that the vlPFC responses we observed rely on explicit occlusion-related signals.

The dependence of vlPFC responses on occlusion level varied across neurons. The responses of some neurons increased gradually with increasing occlusion level whereas the responses of other neurons increased abruptly, even at the lowest occlusion levels. Further experiments are needed to determine whether neuronal sensitivity to occlusion is determined by feedforward inputs, by gating in vlPFC or by the difficulty of the perceptual discrimination.

We propose that vlPFC responses arise from the modulation of occlusion-dependent, shape-selective feedforward signals from V4 by another feedforward signal that is dependent only on the occlusion level. In our simple behavioral task, where the occluding dots have a different color than the occluded shapes, a neuron sensitive to the color and area of the occluding dots could signal the level of occlusion. Indeed, in one monkey performing the same behavioral task used in the current study, we found that the responses of many IT neurons were consistent with encoding the total area of the occluders (*Namima and Pasupathy, 2016*). However, in the natural world, where there are multiple objects and the attributes of the occluders are not known *a priori*, identifying which object is occluded, and by how much, could be challenging. Extending our simple model to tackle more complex, naturalistic cases would likely require the incorporation of attention and memory processes.

## Implications for decision-making and recognition

To perform the sequential shape discrimination task used in the current study, the reference stimulus held in memory must be compared to the test stimulus on the screen. Given its role in working memory, the PFC is a plausible neural locus for this comparison (*Fuster, 1989*; *Kim and Shadlen, 1999*; *Romo and de Lafuente, 2013*). Our results suggest, however, that the comparison of reference and test stimuli is unlikely to be implemented in vlPFC. We found stronger neuronal selectivity in vlPFC for occluded than unoccluded test stimuli. Thus, if behavioral performance depended on comparisons implemented in vlPFC, discriminability would be higher for occluded stimuli and lower for unoccluded stimuli – the opposite of the performance we observed (*Figure 1B*). The weak neuronal responses in vlPFC to unoccluded stimuli are consistent with a report of weak neuronal selectivity in this area for stimulus color in monkeys performing a color change detection task (*Lara and Wallis, 2014*). Together, these findings challenge the notion that vlPFC activity mediates perceptual discriminations of form and color directly. The comparison of sensory representations could be implemented in other parts of the PFC or in sensory cortex, where signals correlated with monkeys' behavioral decisions have been reported (*Kim and Shadlen, 1999*; *Eskandar et al., 1992*; *Miller and Desimone, 1994*; *Wallis and Miller, 2003*; *Romo and Salinas, 2003*; *Zaksas and*

*Pasternak, 2006*; *Kosai et al., 2014*). The evidence for functional connectivity between V4 and lateral PFC during memory maintenance also supports the implementation of decision computations in visual cortex (*Liebe et al., 2012*).

The finding that neuronal responses in vlPFC are stronger for occluded stimuli is consistent with two possibilities—a role for this area in decision-making and a role in the recognition of occluded objects. Given that the monkeys were required to report their perceptual judgments, it is possible that the vlPFC responses we recorded reflect this area's engagement in facilitating decisions under limited sensory evidence. When shape selectivity in visual cortex is weakened by the presence of occlusion (*Kosai et al., 2014*; *Kovács et al., 1995*; *Tang et al., 2014a*), decision-making becomes difficult. In this case, vlPFC feedback might serve to amplify weak signals expressly to facilitate decisions. In this regard, our results are consistent with vlPFC's engagement in tasks of greater difficulty or cognitive demand (*Crittenden and Duncan, 2014*). Importantly, when the task becomes difficult, rather than reflecting task difficulty per se, we propose that vlPFC responses amplify behaviorally relevant signals to facilitate perceptual decisions.

An alternative possibility is that the preference of vlPFC neurons for occluded stimuli may be related to the specific engagement of vlPFC in the recognition of occluded objects. Previous work has argued that the processing of complex visual scenes containing clutter and occlusions may be guided by higher cognitive, memory processes (*Cavanagh, 1991*; *Kveraga et al., 2007*). For example, image representations in early and mid-level areas of the ventral visual pathway may be relayed to higher processing stages where they are compared to stored representations of object prototypes, leading to the recognition of objects in the scene (*Kveraga et al., 2007*). This recognition process may then guide the grouping of appropriate contours and regions, thereby facilitating object segmentation and scene understanding (*McDermott, 2004*; *Kveraga et al., 2007*). Our results are also broadly consistent with the possibility that vlPFC activity embodies a recognition signal that is fed back to V4 to refine object representations. Further experiments are needed to differentiate between the two alternative roles for vlPFC.

## Response dynamics in V4

Studies of object representation and recognition often consider spiking activity only within the first hundred milliseconds after stimulus onset (e.g. *Hung et al., 2005*). The rationale for choosing this early temporal epoch for analysis is based on the argument that successful categorization can be achieved by feedforward processes alone (*VanRullen and Thorpe, 2001*; *Serre et al., 2007*). However, neuronal responses to visual stimuli depend not only on signals carried by feedforward connections but also by feedback and horizontal connections. Feedback and horizontal connections may modulate neuronal responses based on stimulus context and behavioral goals, and confer selectivity for more complex visual stimuli (*Lamme and Roelfsema, 2000*; *Gilbert and Li, 2013*).

The relative contributions of feedforward, feedback and horizontal connections to neuronal responses are hard to disentangle experimentally, but examining the response dynamics provides useful insights. For example, recent studies comparing V4 and V1 response dynamics during contour grouping and scene segmentation tasks suggest that feedback from V4 to V1 enhances the representation of figures and suppresses the representation of backgrounds in V1 (*Chen et al., 2014*; *Poort et al., 2012*). Similarly, our results suggest that feedback from vlPFC to V4 enhances the representation of behaviorally relevant, occluded shapes in V4.

Our model simulations suggest that the enhancement of V4 shape selectivity could be mediated even by weakly tuned vlPFC neurons. This may be because we used only two discriminanda in each experimental session, thereby simplifying the object recognition problem. In this case, any given vlPFC neuron receiving differential input from the two subsets of V4 neurons that signal the two shape discriminanda could contribute to enhanced V4 shape selectivity. We cannot rule out the possibility that IT responses also contribute to V4 responses during the second transient peak. However, our IT recordings suggest this is unlikely because, as in V4, shape selectivity in IT is stronger for unoccluded than occluded stimuli (*Namima and Pasupathy, 2016*). Thus, putative feedback from IT may not be well-suited for enhancing V4 shape selectivity at intermediate occlusion levels.

The V4 neurons in our data set showed a broad range of second peak times and peak magnitudes. The diversity in timing of the second peak is likely because V4 and vlPFC are connected via feedforward and feedback pathways that are direct and indirect, and these different pathways are expected to have different conduction times. It is also possible that feedback signals from vlPFC to

V4 are carried by a sparse projection and then distributed more broadly across V4 via horizontal connections, resulting in longer delays (than expected for disynaptic transmission) between the peak of vlPFC responses and the second peak of V4 responses. The strength of connections between V4 and vlPFC is likely heterogeneous, which could explain the range of second peak magnitudes we documented. Our simulations demonstrate that even weak functional interactions between the two areas could result in a small, second response peak in V4 that may be undetectable in highly variable responses. Overall, the heterogeneous properties of the second response peak support the possibility that V4 neurons with and without two peaks lie along a continuum.

While our model simulations successfully reproduced the diversity of neuronal dynamics observed in V4 in terms of the amplitude and timing of the second transient response peak, this result was achieved by tweaking the parameters of response timing and connection strengths in different instantiations of the two-layer V4–vlPFC interaction model. We do not know whether a large set of interconnected neurons, each with a large number of incoming inputs, could exhibit diverse response dynamics despite differences in response timing and connection strengths. In such a network, 'averaging' across the many inputs each neuron receives could dampen diversity in the response dynamics. It is also possible that in the limiting case where the neural population is large in size, but the number of active incoming connections per neuron is relatively small, substantial variation in response dynamics could persist across neurons. A detailed study of the relationships between network connectivity, network size and neuronal dynamics would be useful to validate the proposed model.

We studied vlPFC and V4 neuronal responses in the same monkeys and using the same behavioral paradigm, thereby facilitating direct comparisons of measurements made in the two cortical areas. Nevertheless, our approach to studying vlPFC contributions to V4 had several limitations. First, we conducted V4 and vlPFC recordings in separate sessions so we cannot compare V4 and vlPFC activity on individual trials. Second, we used only two discriminanda in each behavioral session, so we do not know whether vlPFC neurons are sufficiently sensitive to shape information to mediate the recognition of occluded objects. Third, we did not instruct monkeys to report their behavioral decisions as soon as possible, so we cannot infer the precise epoch of V4 neural activity that mediates perceptual judgments. Specifically, we do not know whether V4 neuronal responses during the second transient peak, which have stronger shape selectivity, contributed to the monkeys' perceptual decisions. Fourth, we do not know whether the engagement of vlPFC neurons in our study was contingent on the monkeys reporting their perceptual decisions, or whether these same neurons would also be engaged in the recognition of partially occluded objects in natural viewing conditions. We hope that future studies will answer these outstanding questions and probe the causal link between V4 and vlPFC activity using perturbations of neuronal activity.

Partial occlusions pose a major challenge to the successful recognition of visual objects because they reduce the evidence available to the brain. Recognizing partially occluded objects could require solving an ill-posed inverse problem (*Helmholtz, 1910*; *Yuille and Kersten, 2006*), one that lacks a unique solution because the retinal image of an occluded object is often compatible with multiple interpretations (e.g. see Bregman's B illusion, *Bregman, 1981*). As a result, recognition must rely not only on information about the physical object but also on information about the occlusion, scene context and perceptual experience. Our results provide support for the hypothesis that feedback signals from vlPFC, which carry information about occlusions, contribute to object representation in V4 and to object recognition under occlusion. Other brain regions, for example IT cortex, are likely to be involved and should be studied. Future experiments are needed to reveal the detailed algorithms used by neurons and circuits to solve object recognition under occlusion.

## Materials and Methods

### Experimental subjects

Two adult male rhesus macaques (*Macaca mulatta*) were prepared for neurophysiological recordings using sterile surgical procedures. For experiments in prefrontal cortex, recording chambers were centered over the principal sulcus and targeted the ventrolateral prefrontal cortex (vlPFC), located ventral to and along the caudal third of the principal sulcus. The stereotaxic, central coordinates of prefrontal recording chambers were derived based on structural MRI images for each animal, and

were ~21 mm anterior of interaural zero and ~19 mm lateral to the midline. For experiments in visual cortex, recording chambers were centered on the dorsal surface along the prelunate gyrus and targeted area V4, extending between the lunate sulcus and the superior temporal sulcus. Recordings from the two areas were carried out serially in the same monkeys, starting with V4 then vlPFC. All animal procedures conformed to NIH guidelines and were approved by the Institutional Animal Care and Use Committee at the University of Washington.

## Neurophysiology

Extracellular recordings were performed using epoxy-coated tungsten microelectrodes (250 µm, FHC) lowered into cortex through an acute microdrive system (Gray Matter Research, 8-channel). Voltage signals were amplified and band-pass filtered (0.1–8 kHz) using a recording system (Plexon Systems, 16-channel). The waveforms of single units were isolated manually using spike-sorting software (Plexon Systems, Offline Sorter). The results reported in the current study are based on 381 vlPFC neurons (260 and 121 from Monkey M and Monkey O, respectively) and on 85 V4 neurons (41 from Monkey M and 44 from Monkey O, respectively). A subset of the V4 neurons (62 of 85 neurons) contributed to a previous study (see *Kosai et al., 2014*).

## Visual stimuli

Visual stimuli were presented on a calibrated CRT monitor (1600 x 1200 pixels; 97 Hz frame rate; 57 cm in front of the monkey). Stimuli were presented against an achromatic gray background of mean luminance 5.4 cd/m$^2$. Stimulus onset and offset times were based on photodiode detection of synchronized pulses in one corner of the monitor. Stimulus presentation and behavioral events were controlled by custom software written in Python (Pype, originally developed by Jack Gallant and James Mazer; *Mazer, 2013*). Eye position was monitored using a 1 kHz infrared eye-tracking system (Eyelink 1000; SR Research).

## Behavioral task

Monkeys performed a sequential shape discrimination task in the presence and absence of occluders (*Figure 1A*). Each trial began with the presentation of a central point (0.1°), which the monkey had to fixate within a circular window of radius 0.75°. After acquiring fixation, two stimuli were presented: a 'reference' stimulus, followed by a 'test' stimulus. The reference stimulus was always an unoccluded, 2D shape. The test stimulus was a 2D shape that was unoccluded or partially occluded by a field of randomly positioned dots. Occlusion level was quantified as the percentage of the shape area that remained visible ('% visible area') and was titrated by varying the diameter of the occluding dots (for details, see *Kosai et al., 2014*). Each stimulus was presented for 600 ms, with an inter-stimulus interval of 200 ms between the reference and the test stimuli. Following a 50 ms delay, the fixation point was extinguished and two peripheral choice targets appeared (left and right dots, 6° eccentric; *Figure 1A*). The monkey reported whether the two shapes presented were the same or different via a saccade to the right or left target, respectively, and within 500 ms of target onset. The monkey received liquid reward for correct performance. In cases where the monkey broke fixation or failed to respond, the trial was repeated later in the session. Behavioral trials were separated by an inter-trial interval of 2 s.

## Approach to data collection

V4 data were collected by studying one neuron at a time and tailoring the shapes, occluding dots, colors and position of the test stimulus to the preferences of the neuron recorded. Based on preliminary characterizations (for details, see *Kosai et al., 2014*) we chose two shapes as the discriminanda: one preferred and one non-preferred. The two shapes were presented in the neuron's preferred color whereas the occluding dots were presented in a contrasting, non-preferred color. The reference stimulus was presented at central fixation whereas the test stimulus was presented at the center of the neuron's RF. The vlPFC data were collected by studying several neurons simultaneously, an approach that precluded tailoring the shapes and occluding dots to the preferences of individual neurons. To equate behavioral task difficulty across V4 and vlPFC recording sessions, we chose, at random, stimulus parameters for each vlPFC recording session from among those used for V4 recording sessions.

Two shapes were used in each behavioral session, yielding four trial conditions per occlusion level (2 shapes x two behavioral outcomes). We studied each neuron's responses to the two shapes under four or more occlusion levels, including the unoccluded case. In V4 recordings, we sampled 4–9 occlusion levels (median = 6). In vlPFC recordings, we sampled 5–6 occlusion levels (median = 5). We only included data from neurons tested with at least seven repeated presentations of each trial condition and of each occlusion level tested. The median number of repeats was 24 for V4 recordings and 15 for vlPFC recordings.

## Data analysis

### Visually responsive and occlusion-sensitive vlPFC neurons

We identified visually responsive vlPFC neurons by comparing the firing rate during a 150 ms window, beginning 80 ms after test stimulus onset, to the firing rate during the fixation epoch before reference stimulus onset. Neuronal responses to the test stimulus were often transient (see *Figure 2*), motivating us to calculate firing rates in a 150 ms window rather than the full duration of stimulus presentation. The 80 ms offset was introduced to account for the visual response latency of neurons. Among 381 vlPFC neurons, 216 (142/260 in Monkey M and 74/121 in Monkey O) were significantly responsive during the test stimulus epoch (t Test, $p<0.01$). All further data analyses were restricted to these visually responsive neurons (57% of vlPFC neurons recorded).

To assess whether neuronal responses to the test stimulus were modulated by shape and/or occlusion level, we conducted a 2-way ANOVA on activity during the same response window defined above, with stimulus shape and occlusion level as factors. Among the 216 visually responsive neurons, the responses of 98 neurons (71/260 in Monkey M and 27/121 in Monkey O) showed a significant dependence on occlusion level ($p<0.05$) and the responses of 66 neurons showed a significant dependence on stimulus shape ($p<0.05$).

### Shape selectivity

To examine the dynamics of neuronal shape selectivity, we performed a sliding-window Receiver Operating Characteristic (ROC) curve analysis on responses to the preferred and non-preferred shapes at each occlusion level. For V4 neurons, the preferred shape was that which evoked the largest average response across all occlusion levels. For vlPFC, because many neurons did not respond to unoccluded shapes, we computed the average response for each shape across all occlusion levels (i.e. visible area <100%) during the test epoch, and identified the preferred shape as that which evoked the largest average response. At every time point (1 ms steps), we counted spikes in a centered window of duration 75 ms (for V4) or 150 ms (for vlPFC). We then assessed shape selectivity by computing the area under the ROC curve derived from the spike count distributions of responses to preferred and non-preferred shapes. Shape selectivity values ranged from 0.5 (unselective) to 1.0 (very selective). To identify the time of maximal shape selectivity for occluded stimuli (*Figure 8D*, red bars), we also computed shape selectivity as described above, pooling across all levels of occlusion tested for each neuron.

### Population response histograms

To generate population response histograms (*Figures 3* and *6*), we normalized the responses of each neuron to the maximum across all occlusion levels then averaged the data at each occlusion level for all neurons. We did not test all neurons at the same occlusion levels (for each neuron, we tested 4–9 occlusion levels), so the number of neurons contributing to the average histograms varied for each occlusion level; these numbers are listed in the figures. For both cortical areas, population response histograms for the occlusion conditions of 44% and 27% visible area were based on only a few neurons and were therefore excluded.

### Peak latency

To find the time to peak response for each neuron, we first constructed an average response histogram from the Gaussian-smoothed ($\sigma = 10$ ms) PSTHs across all occlusion levels. We then identified the time of maximal response between 50–600 ms after test stimulus onset. This temporal window allowed us to identify peaks associated with responses to the test stimulus rather than responses

related to the preceding reference stimulus, memory delay or saccades that followed the test stimulus.

## Peak finding algorithm

To identify V4 neurons with two transient peaks in their responses to occluded shapes, we devised an *ad hoc* algorithm, described below. This procedure was designed to identify neurons with a robust second transient response peak that could not be attributed to small, noisy ripples in the response. For each neuron, we first constructed an average PSTH of its responses to the preferred shape at different occlusion levels, smoothed with a Gaussian function ($\sigma$ = 10 ms). We only included occlusion levels that evoked a response that was at least 33% of the maximal response to the unoccluded preferred shape. We then used a zero-crossing algorithm to identify local peaks within 300 ms of stimulus onset. Small peaks (<50% of the first transient peak) and small trough-to-peak modulation ratios (<15% of local peak magnitude) were rejected as false positives (see *Figure 6—figure supplement 1* for a schematic of the procedure and examples of rejection cases). For each putative peak that met the peak amplitude and modulation criteria, we asked whether there was a statistically significant response increase relative to the preceding trough. To assess statistical significance, we conducted a paired t-Test between single trial spike counts within a 30 ms window centered at the peak and at the preceding trough (p<0.05, Bonferroni corrected). Of 85 V4 neurons, 43 had no robust peaks beyond the first transient that passed the peak amplitude and modulation criteria. Of the remaining 42 neurons, 30 had a second peak that showed a statistically significant response increase relative to the preceding trough; these neurons were classified as having two peaks. Specifically, 29/30 neuons had exactly one peak that qualified as the second response peak.

## Dynamic network model

To evaluate whether interactions between vlPFC and V4 could account for the observed response dynamics, we constructed a network model (*Figure 9*). The model includes two stages of cortical processing that are intended to map onto areas vlPFC and V4. The V4 stage comprises two units ($V4_1$ and $V4_2$), each selective for one of the shapes used in a testing session. The vlPFC stage also comprises two units ($PFC_1$ and $PFC_2$) that receive excitatory feedforward input from V4 units. Each vlPFC unit inherits a preference for stimulus shape from the V4 unit that provides the strongest input (e.g. $PFC_1$ receives the strongest feedforward input from $V4_1$). In the model simulations presented here, feedforward and feedback connections strengths are proportional. However, we have verified that the results hold for a broad range of connection strengths, as long as each vlPFC unit sends stronger feedback input to the V4 unit which provides its dominant feedforward input.

All model parameters (listed in *Table 1*) were chosen to reproduce the response dynamics observed in the experimental data. In addition, we also varied synaptic weights and delays over a range of values (see *Table 2*) to compare the heterogeneity of model units to observed data (*Figure 10—figure supplements 3–5*).

**Table 1.** Parameters used for the model (fixed)

| Parameters | Values |
|---|---|
| $\tau_{V4}(ms)$ | 50 |
| $\tau_{PFC}(ms)$ | 20 |
| $r_{thr,V4}(spikes/s)$ | 20 |
| $r_{thr,PFC}(spikes/s)$ | 0 |
| $\sigma$ | 90 |
| $N$ | 2 |
| $f_{max}$ | 100 |
| $r_{thr1}(spikes/s)$ | 30 |
| $r_{thr2}(spikes/s)$ | 10 |
| $\tau_a(ms)$ | 30 |

DOI: https://doi.org/10.7554/eLife.25784.028

**Table 2.** Parameters used for the model (varied)

| Parameters | Figure 10, Figure 10—figure supplements 1,2 | Figure 10—figure supplement 4 | | | Figure 10—figure supplement 5 | Figure 10—figure supplement 6 |
|---|---|---|---|---|---|---|
| $\tau_{d,ff}\,(ms)$ | 40 | 40 | | | 40 | 40 |
| $\tau_{d,fb}\,(ms)$ | 40 | 40 | | | 20, 40, 80 | 20–80 |
| $w_{\infty,sff}$ | 0.5 | 0.7 | 0.6 | 0.5 | 0.5 | 0.4–0.6 |
| $w_{\infty,wff}$ | 0.2 | 0 | 0.1 | 0.2 | 0.2 | 0–0.2 |
| $w_{\infty,sfb}$ | 1.3 | 1 | | | 0.2, 0.8, 1.3, 1.5 | 0.1–1.3 |
| $w_{\infty,wfb}$ | $w_{\infty,sfb} \cdot w_{\infty,wff}/w_{\infty,sff}$ | | | | | |

DOI: https://doi.org/10.7554/eLife.25784.029

We modeled the dynamical firing rate response of each model V4 unit, $r_{V4,i}$, as:

$$\tau_{V4}\frac{dr_{V4,i}(t)}{dt} = -r_{V4,i}(t) + F\big(U_{V4,i}(t) - r_{thr,V4}\big) + \eta(t),\tag{1}$$

and the firing rate response of each model vlPFC unit, $r_{PFC,\,i}$, as:

$$\tau_{PFC}\frac{dr_{PFC,i}(t)}{dt} = -r_{PFC,i}(t) + \bar{f}_{PFC}\cdot F\big(U_{PFC,i}(t) - r_{thr,PFC}\big) + \eta(t)\tag{2}$$

where $\tau_{V4}$ and $\tau_{PFC}$ denote the time constants of the responses; $t$ denotes time; $F(x)$ is a nonlinear function of the Naka Rushton form given by:

$$F(x) = \begin{cases} \frac{f_{max}\cdot x^N}{\sigma^N + x^N}, & x \geq 0 \\ 0, & x < 0 \end{cases}\tag{3}$$

where $r_{thr,V4}$ and $r_{thr,PFC}$ are the firing rate thresholds, and $\eta$ is a Gaussian white noise term with a standard deviation of $300*\sqrt{dt}$ and a timestep $dt = 0.01\,(ms)$. We omitted the noise term for some simulations (*Figure 10—figure supplements 1–6*). Note that the precise form of the nonlinear function $F(x)$ is not critical; any monotonically increasing nonlinear function with saturation and threshold, along with the dynamics of firing rates defined in *Equations (1) and (2)* provide a standard firing rate model (*Dayan and Abbott, 2005*).

For V4 model units, the input $U_{V4,i}$ was the sum of two sources: (i) excitatory feedforward input from upstream visual areas, $u^i_{FF}$, and (ii) excitatory feedback inputs from vlPFC, $u^i_{PFC(1)}$ and $u^i_{PFC(2)}$. The feedforward input $u^i_{FF}$ confers shape selectivity to the V4 model units and a dependence of their responses on occlusion level (*Figure 9*). For the preferred stimulus, this input is strong and declines gradually with increasing occlusion level. For the non-preferred shape, this input is weak, as is the modulatory influence of occlusion level. The feedforward input, $u^i_{FF}$, was constructed by first convolving a difference of Gaussian filter $k$ (the standard kernel normalized difference of $g_1 = 15 \cdot \exp\left[-\frac{(t-30)^2}{800}\right]$ and $g_2 = 10 \cdot \exp\left[-\frac{(t-50)^2}{800}\right]$)) with a $500ms$-long ramp $R_i(c,t)$ followed by cubing, normalization and half-wave rectification:

$$u^i_{FF} = \left[\frac{(k*R_i)^3}{(\max[k*R_i])^2}\right]_+\tag{4}$$

The ramp function (R), defined separately for the preferred (i = 1) and nonpreferred (i = 2) shapes, increases monotonically with the percentage of visible area (c) and declines over time with a support of 500 ms, that is

$$R_i(c,t) = \begin{cases} (2.5c + 20) - 0.05t, & i = 1, \\ (12c^{1/3} + 120) - 0.05t, & i = 2, \end{cases} \tag{5}$$

when $30 \leq t \leq 530$. $R_i$ is 0 otherwise.

Equations 4 and 5 were designed to simulate the input to V4 units (e.g. *Figure 10A*) with an onset latency of 30 ms, a strong initial transient response, a gradually declining sustained response, collectively lasting ~500 ms. Note that the precise function defining $u_{FF}^i$ is not critical as long as it produces strong input signals for the preferred shape that decrease with increasing occlusion level, thus capturing the observed V4 neuronal response properties.

For the vlPFC units, the input, $U_{PFC,i}$ is the excitatory feedforward inputs from both V4 units, $u_{V4(1)}^i$ and $u_{V4(2)}^i$. In addition, the vlPFC units receive a gain modulation signal, $\overline{f}_{PFC}$, that is proportional to the occlusion level. We modeled $\overline{f}_{PFC}$ as a nonlinear, cubic function of the % visible area, $c$. The function's output was lowest for the unoccluded shape ($c = 100\%$) and increased for higher occlusion levels (*Figure 9*, $\overline{f}$). The coefficients were fit so that the model responses closely resembled the neuronal data, but the qualitative results were independent of the coefficient values used:

$$\overline{f}_{PFC} = -0.0017 \cdot c^3 + 0.39 \cdot c^2 - 29.6 \cdot c + 806. \tag{6}$$

Inputs between model units were modulated by connection weights: $w_{sff}$ for the stronger feedforward inputs from V4 units to vlPFC units of the same shape preference (e.g. V4 unit 1→ vlPFC unit 1), $w_{sfb}$ for the corresponding feedback inputs (e.g. vlPFC unit 1→ V4 unit 1), $w_{wff}$ for the weaker feedforward inputs from V4 units to vlPFC units of a different shape preference (e.g. V4 unit 1→ vlPFC unit 2), and $w_{wfb}$ for the corresponding feedback inputs (e.g. vlPFC unit 1→ V4 unit 2). Thus, the feedback input from vlPFC unit $j$ onto V4 unit $i$, $u_{PFC(j)}^i$, was implemented as follows:

$$u_{PFC(j)}^i(t) = \begin{cases} w_{sfb} \cdot \left[ r_{PFC,j}\left(t - \tau_{d,fb}\right) - r_{thr1} \right]_+, & i = j. \\ w_{wfb} \cdot \left[ r_{PFC,j}\left(t - \tau_{d,fb}\right) - r_{thr1} \right]_+, & i \neq j. \end{cases} \tag{7}$$

where the responses of vlPFC units were thresholded ($r_{thr1}$) and half-wave rectified. This threshold on vlPFC firing rates was introduced to reduce the magnitude of the second transient peak in V4 unit responses to the non-preferred shape (see *Figure 10—figure supplement 2*).

The feedforward excitatory input from V4 unit $j$ to vlPFC unit $i$ was implemented as:

$$u_{V4(j)}^i(t) = \begin{cases} w_{sff} \cdot r_{V4,j}\left(t - \tau_{d,ff}\right), & i = j. \\ w_{wff} \cdot r_{V4,j}\left(t - \tau_{d,ff}\right), & i \neq j. \end{cases} \tag{8}$$

The feedforward and feedback temporal delays between vlPFC and V4 unit responses, $\tau_{d,ff}$ and $\tau_{d,fb}$ were chosen to be consistent with the difference in time between the vlPFC and V4 response peaks observed in our neuronal data.

To prevent the second response peak of V4 units from inducing a second response peak in vlPFC units (see *Figure 10—figure supplement 1B*), the feedforward connections from V4 to vlPFC included an adaptation term, as follows:

$$\begin{aligned} \frac{dw_{ff}}{dt} &= \frac{1}{\tau_a}\left(w_{\infty,ff} - w_{ff}\right) \\ w_{\infty,ff} &\leftarrow 0 \;\; if \;\; r_{PFC,i} \geq r_{thr2} \end{aligned} \tag{9}$$

where the weight $w_{ff}$ of connections from V4 to vlPFC represents both $w_{wff}$ and $w_{sff}$, and evolves with time scale $\tau_a$. When vlPFC activity exceeds the value of $r_{thr2}$ (10 spk/sec, see *Table 1*), the steady state feedforward connection from V4 to vlPFC, $w_{\infty,ff}$, goes to 0, and any subsequent input from V4 will fail to activate vlPFC. The feedback connectivity weight was time-independent and set to steady state values: $w_{sfb} = w_{\infty,sfb}$, $w_{wfb} = w_{\infty,wfb}$.

The set of differential equations, with stochastic noise term $\eta$, was solved using the Forward Euler Method in MATLAB. The initial firing rate values for $r_{V4,i}$ and $r_{PFC,i}$ were set to 0 spikes per second.

The initial connectivity weights were equivalent to the steady-state weights $w_{\infty,sff}$, $w_{\infty,wff}$, $w_{\infty,sfb}$ and $w_{\infty,wfb}$; these and other parameters are given in *Tables 1* and *2*.

The code for the full model is available on Github (*Choi, 2017*). A copy is archived at https://github.com/elifesciences-publications/V4-PFC-dynamics.

## Acknowledgements

We thank Wyeth Bair, Gregory Horwitz and Dina Popovkina for helpful discussions and comments on the manuscript, and Yoshito Kosai for assistance with animal training and V4 data collection. Technical support was provided by the Bioengineering group at the Washington National Primate Research Center. This work was funded by NEI grant R01EY018839 to A Pasupathy, Vision Core grant P30EY01730 to the University of Washington, P51 grant OD010425 to the Washington National Primate Research Center, NSF grant DMS-1056125 to E Shea-Brown, and Washington Research Foundation Innovation Postdoctoral Fellowship in Neuroengineering to H Choi.

## Additional information

### Funding

| Funder | Grant reference number | Author |
| --- | --- | --- |
| Washington Research Foundation | Innovation Postdoctoral Fellowship in Neuroengineering | Hannah Choi |
| National Science Foundation | DMS-1056125 | Eric Shea-Brown |
| National Eye Institute | R01EY018839 | Anitha Pasupathy |
| National Institutes of Health | OD010425 | Anitha Pasupathy |
| National Eye Institute | P30EY01730 | Anitha Pasupathy |

The funders had no role in study design, data collection and interpretation, or the decision to submit the work for publication.

### Author contributions

Amber M Fyall, Data curation, Writing—review and editing; Yasmine El-Shamayleh, Data curation, Writing—original draft, Writing—review and editing; Hannah Choi, Software, Investigation, Methodology, Writing—original draft, Writing—review and editing; Eric Shea-Brown, Conceptualization, Supervision, Writing—review and editing; Anitha Pasupathy, Conceptualization, Resources, Software, Formal analysis, Supervision, Funding acquisition, Validation, Investigation, Visualization, Methodology, Writing—original draft, Project administration, Writing—review and editing

### Author ORCIDs

Yasmine El-Shamayleh (iD) http://orcid.org/0000-0002-5396-2823
Hannah Choi (iD) https://orcid.org/0000-0002-8192-1121
Anitha Pasupathy (iD) http://orcid.org/0000-0003-3808-8063

### Ethics

Animal experimentation: All animal procedures conformed to NIH guidelines and were approved by the Institutional Animal Care and Use Committee at the University of Washington (IACUC Protocol #4133-01).

### Decision letter and Author response

Decision letter https://doi.org/10.7554/eLife.25784.030
Author response https://doi.org/10.7554/eLife.25784.031

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
