## [Decision Letter]

Thank you for submitting your article "Dynamic representation of partially occluded objects in primate prefrontal and visual cortex" for consideration by *eLife*. Your article has been favorably evaluated by David Van Essen (Senior Editor) and three reviewers, one of whom is a member of our Board of Reviewing Editors. The reviewers have opted to remain anonymous.

The reviewers have discussed the reviews with one another and the Reviewing Editor has drafted this letter to crystallize our concerns going forward. We feel the work is important and interesting but key issues remain unresolved that must be addressed satisfactorily to produce an acceptable manuscript.

At this point we are unable to render a binding recommendation and require a response from you indicating the feasibility of your completing the essential tasks in a reasonable period of time – around 2 months. The Board member and reviewers will consider your response and provide a binding decision.

General assessment:

This paper characterizes responses of V4 and vlPFC neurons to partially occluded visual stimuli and suggests that feedback from vlPFC to V4 boosts V4 responses to occluded shapes and helps resolve stimulus identity. The authors recorded vlPFC and V4 neurons from the same monkeys performing the same task, although in separate experimental sessions. They demonstrate that vlPFC neurons respond more strongly and more selectively to occluded stimuli, unlike V4 neurons, which commonly respond most strongly and selectively to unoccluded images. The authors suggest that a subset of V4 neurons respond to occluded images with two distinct peaks (but not all reviewers are convinced that the distinction between subpopulations with one and two peaks are real). The second peak follows the vlPFC response peak and shows similar characteristics to vlPFC. Based on these observations, the authors construct a two-layer neural network in which V4 and vlPFC model units are reciprocally connected and vlPFC responses shape the second peak of V4 units.

The reviewers find the proposal that PFC interacts with V4 to resolve the challenge of solving shape discrimination in the presence of occlusion to be timely and of significant interest. At the same time, the reviewers identified problematic issues with the data analysis that must be resolved before this work could be considered for publication.

Summary:

General concerns about the experimental aspects of the paper include the reproducibility of the main result and the impact of using different approaches when recording from the two brain areas that are compared. General concerns about the model include that it may be unnecessarily complex for the current illustrations provided, and that, even with this complexity, the current illustrations do not reflect population average effects.

Essential revisions:

1) The main claims of the paper rest on the assertion that V4 shape selectivity increases as a function of time. The concerns about this claim are two-fold:

1A) The current illustration is made via an argument that there are two subpopulations of neurons, those that have a second, shape selective peak and those that do not. The reviewers are concerned that the existence of these two subpopulations will not be reproducible. This includes some confusion about the methods that were employed to identify the two peaks, as well as the suspicion that the specific parameters used for this identification were overfit to this particular data set.

The way that the two peaks are identified (subsection “Peak finding algorithm”) is hard to understand: "were within ± 12 ms of at least one third of the peaks identified based on the PSTHs for individual occlusion levels".

How sensitive are these findings to the parameters of the ad hoc peak finding algorithm? Similarly, one may think that the algorithm for detecting double-peak V4 cells has many false positives and the true frequency of cells with two peaks is lower than that suggested in the paper. Can you clarify?

In Figure 6, how can the "Time to peak" for V4 neurons be greater than 300 ms, if the second peak was required to be before 300 ms ("The second peak was constrained to be no later than 300 ms after test stimulus onset")?

The manuscript states: "However, shape selectivity for occluded shapes, particularly at intermediate occlusion levels (visible area 72- 95%), was different for the two groups: neurons with two peaks had significantly greater shape selectivity than neurons without two peaks in the time interval ~200-260 ms after test stimulus onset (t-Test, p < 0.05)." The supporting figure is Figure 8 vs. B. What this is stating exactly? Is the significance tested for each occlusion level separately and it was significant for visible area conditions of 72, 82, 90, and 95%? If so, was Bonferroni correction or another correction applied? How was this time period selected (~200-260 ms)? Was there a correction for testing multiple time periods? Do the results hold in different time periods?

To examine the tuning of the "second peak" the authors use as baseline the activity in an earlier phase of the visual response. Hence, if the neurons respond strongly early on, their response will decrease more strongly during the second peak. In other words, there is suspicion that the results presented in Figure 7 where activity of the second peak decreases if the visible area is large is simply an artifact of this erroneous choice of a time window to compute the baseline. Instead the authors should use the pre-stimulus activity as baseline for all epochs. This choice implies in the subsection “Response change”, in the first equation that *b* depends on *i*, in incoming input. In Figure 7 the y-axis is in units of "normalized response change". How was the normalization?

The average PSTH of V4 subpopulation with two response peaks is quite distinct from the single cell examples. For the population, the second peak is not obvious and responses to unoccluded stimuli stay above the occluded shapes, unlike the single cells in Figure 4 and 5. What causes the discrepancy? Is it the variability of the time of the second peak in V4 neurons? It will be helpful to have a supplementary figure with more single cell examples.

1B) More generally, the claim is that V4 shape selectivity changes as a function of time, and missing is the more direct comparison of shape selectivity for the same neurons early versus later in the response.

2) The broad tuning of neurons in vlPFC seems qualitatively inconsistent with it playing a role in enhancing V4 shape selectivity. The model does not currently resolve this nor is an explanation provided.

3) There are concerns that the functional differences between the responses in PFC and V4 may follow from differences in the experimental paradigm. These include:

3A) What was the effect of tailoring stimuli for neurons in one area and not doing so in another area? Can the latency of responses or their tolerance to occlusion be influenced by tailoring stimuli for neurons?

3B) It is unclear why the authors focus on the vlPFC neurons that are influenced by occlusion. There are also many neurons that do not care about the occlusion and we wonder whether these cells include some neurons that are tuned to the shape of the stimuli. If yes, is their tuning better or worse than that of the neurons that are influence by the occlusion?

What would Figure 3 look like if you included all of the 216 vlPFC neurons that were responsive during the test epoch?

Some cells have a stronger response if the visible area increases. Is it possible that these neurons are better tuned to the shapes than those neurons that are most active for high levels of occlusion?

Related to this last point: is it conceivable that the neurons that increase their response if there is more occlusion are tuned to some aspect of the occluders, e.g. the total surface area or the total perimeter of the occluding dots?

This latter possibility seems to be supported by the finding that the vlPFC preference for occlusion decreased when the occluders had the same color as the background (as is stated in the first paragraph of the subsection “Representation of occluded stimuli in vlPFC”).

There were 381 vlPFC neurons, 98 were significantly modulated by occlusion. How many of these 98 neurons are in monkey M vs. O?

4) Can the model replicate all aspects of the data? Including:

The shape selectivity index in vlPFC is considerably lower than in V4. Can the model in Figure 9 work with reduced shape selectivity?

V4 cells are divided into two subpopulations, one with two peaks and another with a single peak. How can the feedback model explain that many V4 cells do not show two peaks in their responses to occluded stimuli? Are the authors assuming inhomogeneous feedback from vlPFC to V4? Alternatively, they may be suggesting that V4 population includes a continuum of responses that vary from single peak to double peaks and include in-between responses.

Can the model be adjusted to replicate population data in Figure 6? In its current form the model seems to best explain single cell examples. It is unclear how well these examples represent the population.

5) The gain mechanism that the authors propose in their model may be envisioned as PFC receiving two signals – an intertwined shape and occlusion signal and an occlusion-only signal, and then correcting the intertwined estimate with occlusion information. This seems like a bit of a chicken-and-egg problem: how and why would the brain extract occlusion level from occluded stimuli but not shape (preceding the locus at which it disambiguates shape)? Where could this v4-independent but occlusion dependent gain modulation be coming from? The authors suggest IT as a potential source, but IT units receive input from V4 and also feedback to V4. Does the model assume that gain modulation arrives at vlPFC with the same latency as the V4 inputs? Why can't feedback from IT to V4 be the source of the second V4 response peak? The authors cite their SfN abstract for occlusion selective signals in IT. It will be useful to provide more explanation about the results, especially for readers who did not stop by the SfN poster. What if there are two visible stimuli: one that is occluded and one that is not? Would the gain modulation then be different for the two stimuli? What if one stimulus occludes another stimulus?

6) Can the model be simplified? The proposed dynamic model has multiple degrees of freedom and several nonlinearities. Is all this complexity necessary? In the current version of the paper, it is difficult to gain intuition about the model based on the text. Simplifying the model can shine light on which components and nonlinearities are indispensable and therefore reasonable targets for follow-up studies. For example, the adaptation term for V4 projections to vlPFC seems a little arbitrary. Why such an adaptation does not happen for other connections in the model? A similar question can be asked about the half-rectified feedback from vlPFC to V4. Why other connections in the model are not half-rectified? More generally, we encourage the authors to explore the model space a little more extensively for simpler model architectures. As it stands, the network seems like a high-parameter model that one can tweak to get many different types of outputs. Establishing the necessity of the proposed architecture and parameterization requires a little more work.

7) On the interpretation of the effects in V4:

In the first paragraph of the subsection “Representation of occluded stimuli in vlPFC” the authors argue that the responses of some of the vlPFC neurons that are stronger if occlusion is stronger does not depend on the increased task difficulty or attentional demands, but this reasoning is unclear. Furthermore, the authors seem to have changed their mind in the fourth paragraph of the subsection 2Representation of occluded stimuli in vlPFC”, where they argue that vlPFC may amplify weak signals and that vlPFC is engaged in tasks of greater difficulty or cognitive demand.

In the first paragraph of the subsection “Representation of occluded stimuli in vlPFC” the authors argue that because many vlPFC neurons were tuned to shape, that these neurons therefore cannot reflect task difficulty or attentional demands. This argument does not hold because neurons may well be tuned to multiple aspects of a task.

8) How did you know whether penetrations were indeed in vlPFC? Was histology performed?

9) [Additional comment sent to the authors in response to authors’ plan for revision]: The authors should clarify if there is a real dichotomy between neurons with one versus two peaks, as well as how broad the distribution of the timing for the second peak is. Can they really convince the reader that they are not simply amplifying noise with their analysis? Furthermore, they now make the point that the tuning is stronger during the second peak. We would like to know if this is not simply predicted by the presence of extra spikes – i.e. the presence of a peak implies some extra spikes at a certain point in time.

[Editors' note: further revisions were requested prior to acceptance, as described below.]

Thank you for resubmitting your work entitled "Dynamic representation of partially occluded objects in primate prefrontal and visual cortex" for further consideration at *eLife*. Your revised article has been favorably evaluated by David Van Essen (Senior Editor), a Reviewing Editor, and two reviewers.

The manuscript has been improved but there are some remaining issues that need to be addressed before acceptance, as described below by reviewer #3. We envision that these revisions will be straightforward to carry out and that verification can be handled by the Reviewing Editor.

*Reviewer #2:*

The authors have addressed the most critical comments. There are structural weaknesses in the dataset that keep alternative interpretations plausible. However, I believe the authors' interpretation is strengthened by the new analyses and the paper passes threshold for publication.

*Reviewer #3:*

In their revision, Fyall et al. have addressed many of my concerns satisfactorily. They have made it clearer that some of the V4 neurons have a second peak that is more prominent in case of occlusion (Figure 4 represents a compelling example). Also, the peak detection method is now more convincing and it is also better documented. It remains unclear whether activity in vlPFC indeed contributes to late V4 activity and it is therefore conceivable that there are additional areas that could contribute to late V4 activity. Yet, I do realize that demonstrating the causal link between dlPFC and V4 would require a different approach, which would be beyond the scope of the present contribution. However, establishing such a causal link might be an important topic for future research, and the authors could mention this point, which could be added to the paragraph of suggested future work (subsection “Response dynamics in V4”, fifth paragraph).

Remaining points:

1) I find it difficult to understand why the vlPFC neurons do not respond so well when the occluders have the same color of the background (subsection “Representation of occluded stimuli in vlPFC”, second paragraph). I would suspect that the processes for shape recognition would remain the same. Or did the monkeys' performance show signs that this was not the case?

2) Quite some p-values are lacking, three examples:

"The responses of most of these occlusion-sensitive 155 neurons (71/98) increased with increasing occlusion level".

"Even for the small subset of vlPFC neurons that responded more strongly to unoccluded stimuli (27/98), shape selectivity was not stronger for unoccluded than occluded stimuli (see Figure 3—figure supplement 2)."

"Shape selectivity for occluded shapes was significantly higher during the second peak than during the first peak."

3) "Of 85 neurons, 30 neurons (~35%) were classified as having two peaks". How were these cells distributed across the two monkeys?

4) The model with interactions between vlPFC and V4 seems still somewhat simplistic as there are only a few neurons and the variation (in effect size and timing) across neurons shown in the figures is actually a variation across neurons in different models rather than a variation of neurons within the same model. In networks with many units and reciprocal connections, the network dynamics might actually work against variation across neurons. The authors should discuss this. It would be great if it would be possible to show the same range of differences between neurons within a same model, but I will not insist on such a demonstration given that making such a larger model might require a substantial investment of time.

5) "We cannot rule out the possibility that IT responses also contribute to V4 responses during the second transient peak. However, our IT recordings suggest this is unlikely because, as in V4, shape selectivity in IT is stronger for unoccluded than occluded stimuli". Is it conceivable that some IT neurons also have two phases in their response where the second phase is more pronounced in the presence of occlusion? It would be great if the authors could look for this possibility in the previous data set by Namima and Pasupathy, 2016? If the two phases are there it would strengthen the paper, but it would also be interesting if that is not the case.

6) Equations 4/5: I failed to see the logic of these equations, would it be possible to clarify this? Equation 9: what is *_thr2_*?

7) I found Figure 8—figure supplement 6 confusing: how do you compute y/z for neurons with one peak?

---

## [Author Response]

*Essential revisions:*

*1) The main claims of the paper rest on the assertion that V4 shape selectivity increases as a function of time. The concerns about this claim are two-fold:*

*1A) The current illustration is made via an argument that there are two subpopulations of neurons, those that have a second, shape selective peak and those that do not. The reviewers are concerned that the existence of these two subpopulations will not be reproducible. This includes some confusion about the methods that were employed to identify the two peaks, as well as the suspicion that the specific parameters used for this identification were overfit to this particular data set.*

*The way that the two peaks are identified (subsection “Peak finding algorithm”) is hard to understand: "were within ± 12 ms of at least one third of the peaks identified based on the PSTHs for individual occlusion levels".*

*How sensitive are these findings to the parameters of the ad hoc peak finding algorithm? Similarly, one may think that the algorithm for detecting double-peak V4 cells has many false positives and the true frequency of cells with two peaks is lower than that suggested in the paper. Can you clarify?*

We appreciate the reviewers’ general concern regarding the reproducibility of our finding that some V4 neurons have two transient response peaks. We address this concern in our revision in several ways:

i) We have revised and simplified the algorithm employed for peak finding, and we have added a new supplementary figure to explain this procedure schematically (Figure 6—figure supplement 1). Specifically, the steps identified as confusing above have now been removed and replaced with a statistical test (see below).

ii) We have repeated our analyses using different parameter choices, and have added two supplementary figures (Figure 8—figure supplement 3 and Figure 8—figure supplement 4) demonstrating the robustness of the main findings described in the manuscript (Figure 6 and Figure 8) across different parameters.

iii) To mitigate concerns about false positives, we have added a statistical criterion to our peak finding algorithm. Only those neurons with a statistically significant (paired t-Test, p < 0.05, Bonferroni corrected) increase in response at the time of the putative second peak are categorized as having two peaks.

iv) We include an independent, model-based procedure for identifying neurons with two transient response peaks that does not rely on the ad hoc peak-finding algorithm. Briefly, this model-based procedure detects neurons for which the response PSTHs for occluded stimuli cannot be modeled as a linear scaling of the response PSTH for unoccluded stimuli. We present two new supplementary figures (Figure 8—figure supplement 5 and Figure 8—figure supplement 6) demonstrating a good correspondence between the results of this model-based procedure and those of the ad hoc peak-finding procedure used previously.

*In Figure 6, how can the "Time to peak" for V4 neurons be greater than 300 ms, if the second peak was required to be before 300 ms ("The second peak was constrained to be no later than 300 ms after test stimulus onset")?*

We have corrected this plotting error.

*The manuscript states: "However, shape selectivity for occluded shapes, particularly at intermediate occlusion levels (visible area 72- 95%), was different for the two groups: neurons with two peaks had significantly greater shape selectivity than neurons without two peaks in the time interval ~200-260 ms after test stimulus onset (t-Test, p < 0.05)." The supporting figure is Figure 8 vs. B. What this is stating exactly? Is the significance tested for each occlusion level separately and it was significant for visible area conditions of 72, 82, 90, and 95%? If so, was Bonferroni correction or another correction applied? How was this time period selected (~200-260 ms)? Was there a correction for testing multiple time periods? Do the results hold in different time periods?*

For improved clarity, we have revised this analysis and we present the new results in the revised manuscript. We now use two t Tests to ask whether shape selectivity was significantly stronger for neurons with two peaks than neurons without two peaks at two time points: one centered around the first peak (69–99 ms) and another centered around the second peak (199–229 ms) of V4 responses. For occluded stimuli, shape selectivity was significantly stronger for neurons with two peaks during the second peak but not during the first peak. For unoccluded stimuli, shape selectivity was not significantly different between the two neuronal subsets during either time point.

*To examine the tuning of the "second peak" the authors use as baseline the activity in an earlier phase of the visual response. Hence, if the neurons respond strongly early on, their response will decrease more strongly during the second peak. In other words, there is suspicion that the results presented in Figure 7 where activity of the second peak decreases if the visible area is large is simply an artifact of this erroneous choice of a time window to compute the baseline. Instead the authors should use the pre-stimulus activity as baseline for all epochs. This choice implies in the subsection “Response change”, in the first equation that b depends on i, in incoming input. In Figure 7 the y-axis is in units of "normalized response change". How was the normalization?*

As requested by the reviewers, we present average activity plots with respect to a single prestimulus baseline for both epochs in the revised manuscript (Figure 7). The results show clearly that V4 neuronal sensitivity to occlusion level changes over time. We hope that this revised analysis of the data mitigates the reviewers’ concerns regarding the time window over which baseline activity is computed. Additionally, we clarify how the response normalization was implemented in the Materials and methods section and in the figure legend. Briefly, prior to constructing the population average, each neuron’s PSTHs were normalized to the maximum value across time and occlusion level. This strategy ensured that differences in peak firing rates between neurons did not impact the population averages.

*The average PSTH of V4 subpopulation with two response peaks is quite distinct from the single cell examples. For the population, the second peak is not obvious and responses to unoccluded stimuli stay above the occluded shapes, unlike the single cells in Figure 4 and 5. What causes the discrepancy? Is it the variability of the time of the second peak in V4 neurons? It will be helpful to have a supplementary figure with more single cell examples.*

We clarify this point in the revised Results section. We agree with the reviewers that the second response peak is less obvious in the population response averages than in the responses of individual example neurons. Our new model simulations (Figure 10—figure supplement 6) verify that this difference could arise due to variability in the timing of the second peak across V4 neurons. As requested by the reviewers, we include a new supplementary figure with additional example V4 neuronal responses from six neurons with two transient response peaks (Figure 5—figure supplement 1). We would like to note that for many neurons with two peaks, the responses to occluded stimuli do not exceed responses to unoccluded stimuli during the second peak (e.g. see Figure 5—figure supplement 1). Rather, our results suggest that the difference in responses to occluded stimuli is larger than that for unoccluded stimuli during the second peak (Figure 7).

*1B) More generally, the claim is that V4 shape selectivity changes as a function of time, and missing is the more direct comparison of shape selectivity for the same neurons early versus later in the response.*

As requested by the reviewers, we have added a new scatter plot providing a direct comparison of shape selectivity for occluded stimuli during the first and second peak for V4 neurons with two peaks (Figure 8). We describe this analysis in the revised Results section. Shape selectivity for occluded stimuli was significantly stronger during the second peak than the first peak. In contrast, we observed no significant difference in shape selectivity for V4 neurons without two peaks. We also observed no significant difference in shape selectivity for unoccluded stimuli for V4 neurons with or without two peaks.

*2) The broad tuning of neurons in vlPFC seems qualitatively inconsistent with it playing a role in enhancing V4 shape selectivity. The model does not currently resolve this nor is an explanation provided.*

We address this point in the revised Discussion section. Our new model simulations demonstrate that feedback signals from weakly shape-selective vlPFC neurons could enhance shape selectivity in V4 (Figure 10—figure supplement 4). In the behavioral task used, the occluded shape used in each experimental session is restricted to one of two choices, thus simplifying the object recognition problem. In this case, any given vlPFC neuron receiving differential input from V4 neurons that signal the two shape discriminanda could contribute to enhanced V4 shape selectivity. We would need additional experiments to determine how vlPFC and V4 might interact in more general situations. It is likely that recognition and memory-related mechanisms would contribute.

*3) There are concerns that the functional differences between the responses in PFC and V4 may follow from differences in the experimental paradigm. These include:*

*3A) What was the effect of tailoring stimuli for neurons in one area and not doing so in another area? Can the latency of responses or their tolerance to occlusion be influenced by tailoring stimuli for neurons?*

The lack of stimulus tailoring cannot explain away our two major vlPFC findings: i) the large fraction of vlPFC neurons that respond preferentially to occluded stimuli, ii) the stronger shape selectivity under occlusion. We address this point extensively in the revised Discussion section.

Tailoring stimuli in V4 may explain the preponderance of neurons in our V4 dataset that responded preferentially to unoccluded shapes. Without tailoring stimuli in vlPFC, we would expect a roughly equal proportion of neurons showing responses that increased and decreased with increasing occlusion level. However, we found that 72% of vlPFC neurons responded preferentially to occluded stimuli, a proportion that deviates significantly from the null hypothesis (binomial test, p < 0.01).

Furthermore, because the visible difference between any two shapes declines with increasing occlusion, we expect shape selectivity to decline with increasing occlusion regardless of whether we tailored the stimuli to each neuron. Stronger shape selectivity in the vlPFC under occlusion defies this expectation. We do not expect that stimulus tailoring (or lack thereof) influenced neuronal response latency for two reasons. First, in vlPFC, we did not find a statistically significant difference in response latency between neurons that were shape-selective (N=66) and neurons that were visually responsive but not shape-selective (N=150) (t Test, p = 0.55). Second, as part of an ongoing study in the lab, we find that V4 response latencies are similar for preferred and non-preferred stimuli (Zamarashkina et al., VSS Abstracts 2017).

*3B) It is unclear why the authors focus on the vlPFC neurons that are influenced by occlusion. There are also many neurons that do not care about the occlusion and we wonder whether these cells include some neurons that are tuned to the shape of the stimuli. If yes, is their tuning better or worse than that of the neurons that are influence by the occlusion?*

We address this point with a new supplementary figure (Figure 3—figure supplement 1), and revised text in the Results section. Briefly, of 216 visually-responsive vlPFC neurons, 66 were shape selective and, of these, 41 were occlusion-sensitive (two-way ANOVA, p < 0.05). Thus, few neurons (25/216) were classified as shape-selective but not occlusion-sensitive by our statistical tests. We now show average shape selectivity for four different vlPFC neuronal subsets (Figure 3—figure supplement 1): (1) shape-selective but not occlusion-sensitive neurons (N=25), (2) all shape-selective neurons (N=66), (3) neurons that carry task relevant information, i.e. either shape-selective or occlusion sensitive (N=123), (4) all visually-responsive neurons (N=216). Shape selectivity was stronger for shape selective neurons than occlusion-sensitive neurons. Notably, however, shape selectivity was strongest at intermediate occlusion levels, for both types of neurons.

*What would Figure 3 look like if you included all of the 216 vlPFC neurons that were responsive during the test epoch?*

As requested by the reviewers, we present these data in a new supplementary figure (Figure 3—figure supplement 1). Compared to Figure 3, the overall magnitude of shape selectivity is reduced due to the inclusion of more neurons, but otherwise the trends are similar. Specifically, shape selectivity is strongest for stimuli at intermediate occlusion levels in both figures.

*Some cells have a stronger response if the visible area increases. Is it possible that these neurons are better tuned to the shapes than those neurons that are most active for high levels of occlusion?*

We address this point with a new supplementary figure (Figure 3—figure supplement 2) and revised text in the Results section. This figure shows shape selectivity across occlusion level separately for vlPFC neurons that prefer lower occlusion levels (higher% visible area, Figure 3—figure supplement 2) and for neurons that prefer higher occlusion levels (lower% visible area, Figure 3—figure supplement 2). The data are more variable in A due to the inclusion of fewer neurons. Nevertheless, two patterns are evident: (1) even for neurons that respond more strongly to unoccluded stimuli (Figure 3—figure supplement 2), shape selectivity is not stronger for unoccluded than occluded stimuli (compare black and colored lines); (2) neurons that prefer higher occlusion levels are more shape-selective under occlusion (compare colors in Figure 3—figure supplement 2 versus Figure 3—figure supplement 2).

*Related to this last point: is it conceivable that the neurons that increase their response if there is more occlusion are tuned to some aspect of the occluders, e.g. the total surface area or the total perimeter of the occluding dots?*

*This latter possibility seems to be supported by the finding that the vlPFC preference for occlusion decreased when the occluders had the same color as the background (as is stated in the first paragraph of the subsection “Representation of occluded stimuli in vlPFC”).*

We now clarify this point in the revised Discussion section. vlPFC neurons that prefer higher occlusion levels may be sensitive to the total area of the occluding dots. This possibility is consistent with the finding that neuronal preference for the occluding dots is reduced when these dots are rendered in the same color as the background. However, it is important to note that many occlusion-sensitive vlPFC neurons are also selective for the occluded shape, so their responses reflect the characteristics of both the occluding dots and the occluded shape.

*There were 381 vlPFC neurons, 98 were significantly modulated by occlusion. How many of these 98 neurons are in monkey M vs. O?*

We include these numbers in the revised Materials and methods section. In monkey M, 71/260 neurons (27%) were occlusion-sensitive. In monkey O, 27/121 neurons (22%) were occlusion-sensitive.

*4) Can the model replicate all aspects of the data? Including:*

*The shape selectivity index in vlPFC is considerably lower than in V4. Can the model in Figure 9 work with reduced shape selectivity?*

We have revised our model in response to this question and verified that the model can produce vlPFC unit responses with different magnitudes of shape selectivity. Specifically, each vlPFC model unit now receives input from both V4 units and the relative connection strengths from V4 to vlPFC units govern the strength of shape selectivity in the vlPFC unit. Figure 10—figure supplement 4 demonstrates that our model captures the overall trends observed in the neuronal data (i.e. two transient response peaks in V4, and a preference for occluded stimuli in vlPFC) under a wide range of vlPFC shape selectivity magnitudes.

*V4 cells are divided into two subpopulations, one with two peaks and another with a single peak. How can the feedback model explain that many V4 cells do not show two peaks in their responses to occluded stimuli? Are the authors assuming inhomogeneous feedback from vlPFC to V4? Alternatively, they may be suggesting that V4 population includes a continuum of responses that vary from single peak to double peaks and include in-between responses.*

We address this question with a supplementary figure (Figure 10—figure supplement 5) and new text in the Discussion section. Figure 10—figure supplement 5 demonstrates that, by varying the strength of feedback connections from V4 to vlPFC, the model can produce V4 unit response dynamics that vary along a continuum, with some units having one peak and others having two peaks. Together with the broad range of second peak magnitudes observed in our neuronal data, these model simulations support the hypothesis that V4 neurons with and without two peaks form a continuum. It is also possible that projections from vlPFC to V4 are inhomogeneous. Anatomical estimates of the proportion of V4 neurons that receive projections from vlPFC and quantification of the diversity and strength of these projections would be needed to fully resolve this question.

*Can the model be adjusted to replicate population data in Figure 6? In its current form the model seems to best explain single cell examples. It is unclear how well these examples represent the population.*

We address this question in two supplementary figures (Figure 10—figure supplement 5 and Figure 10—figure supplement 6). Figure 10—figure supplement 5 demonstrates that, by varying the synaptic delays and strengths in the feedback from vlPFC to V4, the model can produce a variety of V4 responses, with second response peaks that have a range of magnitudes and peak times. Figure 10—figure supplement 6 demonstrates that a simulated population average of 258 V4 model units produced a population-level PSTH that resembled the neuronal data.

*5) The gain mechanism that the authors propose in their model may be envisioned as PFC receiving two signals – an intertwined shape and occlusion signal and an occlusion-only signal, and then correcting the intertwined estimate with occlusion information. This seems like a bit of a chicken-and-egg problem: how and why would the brain extract occlusion level from occluded stimuli but not shape (preceding the locus at which it disambiguates shape)?*

We expand on this point in the revised Discussion section. In our behavioral task, the occluding dots and occluded shapes have different colors. In this simple case, extracting information about occlusion level is easy, and can be mediated by a neuron sensitive to the total area and color of the occluding dots. In the more complex case of natural vision, extracting information about occlusion level could be substantially harder. Future experiments in which features of occluding objects (e.g. color, shape and size) are varied across trials will be needed to extend our findings to more naturalistic cases.

Extracting information about the occluded shape is hard even for our simple task because occlusion makes parts of the shape inaccessible to the visual system. So, a neuron sensitive to boundary form or to the area rendered in a specific color will show reduced responses under occlusion and thus can only provide an intertwined shape and occlusion signal.

*Where could this v4-independent but occlusion dependent gain modulation be coming from? The authors suggest IT as a potential source, but IT units receive input from V4 and also feedback to V4. Does the model assume that gain modulation arrives at vlPFC with the same latency as the V4 inputs? Why can't feedback from IT to V4 be the source of the second V4 response peak?*

We address these questions with a supplementary figure (Figure 10—figure supplement 3) and new text in the revised Results and Discussion sections. It is possible that occlusion-only signals arise first at the level of V4 or, alternatively, IT. Future experiments comparing the preponderance and latency of occlusion-only signals in the two cortical areas are needed to uncover their likely origin. We have now implemented changes to the model to allow for the occlusion signals to arrive either from V4 or from IT cortex (Figure 10—figure supplement 3) by the varying the delay in the time of arrival of the gain modulation signal relative to shape sensitive signals.

We cannot rule out the possibility that IT responses contribute to V4 responses during the second transient peak. However, our IT recordings (Namima and Pasupathy, 2016) suggest this is unlikely because, as in V4, shape selectivity in IT is stronger for unoccluded than occluded stimuli. Thus, putative feedback from IT may not be well-suited for enhancing V4 shape selectivity at intermediate occlusion levels.

*The authors cite their SfN abstract for occlusion selective signals in IT. It will be useful to provide more explanation about the results, especially for readers who did not stop by the SfN poster.*

As requested by the reviewers, we expand on this point in the revised Discussion section. Briefly, recordings performed in IT cortex of one monkey performing the same behavioral task used in the current study show that the responses of many IT neurons are consistent with encoding the total area of the occluding dots (Namima and Pasupathy, 2016).

*What if there are two visible stimuli: one that is occluded and one that is not? Would the gain modulation then be different for the two stimuli? What if one stimulus occludes another stimulus?*

We speculate on these points briefly in the revised Discussion section, although these questions are best addressed with new experiments. When multiple stimuli are presented within a V4 neuron’s receptive field, past studies (e.g. Moran and Desimone, 1985) suggest that neuronal responses are dictated largely by the attended stimulus. Extending our results to the case of two stimuli, one occluded and one unoccluded, we expect gain modulation to be low or high depending on whether the unoccluded or the occluded stimulus is the attended, behaviorally-relevant stimulus.

When one stimulus occludes another and the attributes of the occluders are not known a priori, identifying which object is occluded, and by how much, could be challenging. To extend our simple model to tackle more complex, naturalistic cases would likely require the incorporation of attention and memory processes.

*6) Can the model be simplified? The proposed dynamic model has multiple degrees of freedom and several nonlinearities. Is all this complexity necessary? In the current version of the paper, it is difficult to gain intuition about the model based on the text. Simplifying the model can shine light on which components and nonlinearities are indispensable and therefore reasonable targets for follow-up studies. For example, the adaptation term for V4 projections to vlPFC seems a little arbitrary. Why such an adaptation does not happen for other connections in the model? A similar question can be asked about the half-rectified feedback from vlPFC to V4. Why other connections in the model are not half-rectified? More generally, we encourage the authors to explore the model space a little more extensively for simpler model architectures. As it stands, the network seems like a high-parameter model that one can tweak to get many different types of outputs. Establishing the necessity of the proposed architecture and parameterization requires a little more work.*

We address this question in the revised manuscript with six supplementary figures (Figure 10—figure supplement 1–Figure 10—figure supplement 6) and two new Results sections titled, “Parsimonious model construction” and “Heterogeneity in V4 and vlPFC responses”.

We demonstrate the necessity of four mechanisms included in the model: i) feedback from vlPFC to V4, ii) synaptic adaptation in the feedforward inputs from V4 to vlPFC, iii) a half-wave rectifying nonlinearity on feedback signals from vlPFC to V4, and, iv) gain modulation signals on vlPFC. First, a network with only feedforward connections from V4 to vlPFC units fails to generate a second response peak in V4 units (Figure 10—figure supplement 1). Second, without synaptic adaptation on the feedforward connections from V4 to vlPFC, the feedforward-feedback loop positively reinforces neuronal activity in V4 and PFC units (Figure 10—figure supplement 1), resulting in “ringing” and “blow up” of responses, which is inconsistent with the neuronal data. Third, without half-wave rectification on feedback signals from vlPFC units to V4 units, the model produces a large second peak even in V4 responses to non-preferred stimuli (Figure 10—figure supplement 2), which is inconsistent with the neuronal data (Figure 8—figure supplement 2). Given that such nonlinearities occur at synapses, our model suggests that feedback from vlPFC may arrive in V4 di-synaptically. Finally, we also illustrate the necessity of the occlusion-dependent gain modulation on vlPFC, without which vlPFC unit responses decrease with increasing occlusion (Figure 10—figure supplement 1). We also repeated our model simulations for a range of parameter values associated with V4–vlPFC feedforward connection strengths (Figure 10—figure supplement 4), feedback synaptic strengths and delays (Figure 10—figure supplement 5 and Figure 10—figure supplement 6) and the timing of gain modulation (Figure 10—figure supplement 3). We hope these results validate the necessity of the proposed model architecture and parameter regimes.

*7) On the interpretation of the effects in V4:*

*In the first paragraph of the subsection “Representation of occluded stimuli in vlPFC” the authors argue that the responses of some of the vlPFC neurons that are stronger if occlusion is stronger does not depend on the increased task difficulty or attentional demands, but this reasoning is unclear. Furthermore, the authors seem to have changed their mind in the fourth paragraph of the subsection 2Representation of occluded stimuli in vlPFC”, where they argue that vlPFC may amplify weak signals and that vlPFC is engaged in tasks of greater difficulty or cognitive demand.*

We have revised relevant portions of the text to clarify our points:

“Given that many vlPFC neurons are selective for the shape of the occluded stimulus, it is unlikely that the observed vlPFC responses solely reflect task difficulty level or attentional demands. If difficulty or attention could fully explain vlPFC responses, then occlusion sensitive neurons would not be shape-selective (i.e. the PSTHs in Figure 2 and Figure 2 would be identical).”

“Our results are consistent with vlPFC’s engagement in tasks of greater difficulty or cognitive demand (Crittenden and Duncan, 2014). Importantly, when the task becomes difficult, rather than reflecting difficulty per se, we propose that vlPFC responses amplify behaviorally relevant signals to facilitate perceptual decisions.”

*In the first paragraph of the subsection “Representation of occluded stimuli in vlPFC” the authors argue that because many vlPFC neurons were tuned to shape, that these neurons therefore cannot reflect task difficulty or attentional demands. This argument does not hold because neurons may well be tuned to multiple aspects of a task.*

We have revised the text to clarify our point (see our responses to the previous point). The reviewers are correct in that vlPFC neurons can be tuned to multiple aspects of a task. Our assertion is simply that the responses of vlPFC neurons in our dataset cannot solely reflect attentional demands or task difficulty because these responses also signal the identity of the occluded shape.

*8) How did you know whether penetrations were indeed in vlPFC? Was histology performed?*

We have added text to the Materials and methods section to address this question. Briefly, we conduced structural MRIs for each monkey and localized the principal sulcus and arcuate sulcus based on these data. We then positioned vlPFC recording chambers based on these stereotactic coordinates, centered roughly at 21 mm anterior of interaural zero and 19 mm lateral to the midline.

With regards to histological confirmation, the monkeys are currently contributing to neurophysiological studies of IT, thus precluding histological analysis.

*9) [Additional comment sent to the authors in response to authors’ plan for revision]: The authors should clarify if there is a real dichotomy between neurons with one versus two peaks, as well as how broad the distribution of the timing for the second peak is. Can they really convince the reader that they are not simply amplifying noise with their analysis?*

We address this concern with several new analyses and supplementary figures (Figure 8—figure supplement 5 and Figure 8—figure supplement 6). We summarize our approach to these manuscript revisions below.

i) Our revised peak-finding algorithm now includes a statistical test. Only those neurons that show a statistically significant increase in response around the time of a putative second peak are classified as having two peaks.

ii) To further guard against false positives, we present a new, model-based procedure for identifying neurons with two peaks that is independent of the ad hoc peak-finding algorithm. We show that this model-based procedure works well for example neurons (Figure 8—figure supplement 5) and that the results generated show good correspondence with the results from the revised peak-finding algorithm (Figure 8—figure supplement 6).

iii) Rather than a complete dichotomy, we propose a continuum of functional response properties, akin to the continuum of V1 simple and complex cells. The strength of connections between V4 and vlPFC are likely to be heterogeneous, and our model simulations demonstrate that even weak functional interactions between the two areas could result in a small, second response peak in V4 that may be undetectable in highly variable responses (Figure 10—figure supplement 4 and Figure 10—figure supplement 5).

iv) With regards the timing of the second peak, we designed our peak-detection algorithm to identify response transients within a 300 ms window from test stimulus onset. This choice of temporal window was motivated by the fact that a large fraction of vlPFC neurons responded within 150ms of test stimulus onset. No additional neurons passed the criterion for significance when we extended this window to 500 ms (to exclude responses associated with the offset of the test stimulus at 600 ms). Thus, the distribution of second peak times shown (Figure 6) cannot be attributed to our choice of temporal window.

Furthermore, they now make the point that the tuning is stronger during the second peak. We would like to know if this is not simply predicted by the presence of extra spikes – i.e. the presence of a peak implies some extra spikes at a certain point in time.

We address this concern with a new analysis and a supplementary figure (Figure 8—figure supplement 2).Shape selectivity for occluded stimuli was stronger at the time of the second peak even for neurons with two peaks whose responses to the preferred shape were stronger during the first peak than the second peak (Figure 8—figure supplement 2). Thus, extra spikes at the time of the second peak cannot entirely predict stronger shape selectivity during this epoch. For neurons that showed stronger shape selectivity during the second peak, the magnitude of the second peak relative to the first was larger for the preferred shape than the non-preferred shape (Figure 8—figure supplement 2). We argue that this differential enhancement of responses to the preferred shape serves to amplify shape selectivity.

[Editors' note: further revisions were requested prior to acceptance, as described below.]

*Reviewer #3:*

*In their revision, Fyall et al. have addressed many of my concerns satisfactorily. They have made it clearer that some of the V4 neurons have a second peak that is more prominent in case of occlusion (Figure 4 represents a compelling example). Also, the peak detection method is now more convincing and it is also better documented. It remains unclear whether activity in vlPFC indeed contributes to late V4 activity and it is therefore conceivable that there are additional areas that could contribute to late V4 activity. Yet, I do realize that demonstrating the causal link between dlPFC and V4 would require a different approach, which would be beyond the scope of the present contribution. However, establishing such a causal link might be an important topic for future research, and the authors could mention this point, which could be added to the paragraph of suggested future work (subsection “Response dynamics in V4”, fifth paragraph).*

We thank the reviewer for raising this point, which we have now incorporated into the Discussion.

*Remaining points:*

*1) I find it difficult to understand why the vlPFC neurons do not respond so well when the occluders have the same color of the background (subsection “Representation of occluded stimuli in vlPFC”, second paragraph). I would suspect that the processes for shape recognition would remain the same. Or did the monkeys' performance show signs that this was not the case?*

We did not observe a difference in behavioral performance (% correct) when the occluders were in the same color as the background. However, given that we used a fixed duration task design, we do not know if behavioral reaction times might have been affected by the color of the occluders. The stronger vlPFC responses we recorded when the occluders were in a contrasting color supports our hypothesis that vlPFC responses arise from the modulation of occlusion-dependent, shape-selective feedforward signals from V4 by another feedforward signal that is dependent only on occlusion level. The latter signal may be weaker when the occluders were in the same color as the background, thus yielding weaker vlPFC responses.

*2) Quite some p-values are lacking, three examples:*

*"The responses of most of these occlusion-sensitive 155 neurons (71/98) increased with increasing occlusion level".*

*"Even for the small subset of vlPFC neurons that responded more strongly to unoccluded stimuli (27/98), shape selectivity was not stronger for unoccluded than occluded stimuli (see Figure 3—figure supplement 2)."*

We have re-written the relevant text to clarify this analysis, and to provide the results of significance testing for these two instances pointed out by the reviewer.

Neurons that were classified as occlusion-sensitive based on a two-way ANOVA (p < 0.05) were divided into two groups based on the sign of the regression slope between occlusion level and responses: 71 neurons had a negative slope, indicating that their responses were stronger for occluded stimuli, whereas 27 neurons had a positive or zero slope, indicating that their responses were strongest or comparable for unoccluded stimuli. Of the 98 occlusion sensitive neurons, 59 neurons had a slope that was significantly less than zero whereas 17 neurons had a slope that was significantly greater than zero (p < 0.05).

*"Shape selectivity for occluded shapes was significantly higher during the second peak than during the first peak."*

We have added the significance level to the end of the sentence (t Test, p < 0.01).

*3) "Of 85 neurons, 30 neurons (~35%) were classified as having two peaks". How were these cells distributed across the two monkeys?*

Of the 30 neurons classified as having two peaks, 14 were recorded in monkey O and 16 were recorded in monkey M. We include this information in the revised manuscript.

*4) The model with interactions between vlPFC and V4 seems still somewhat simplistic as there are only a few neurons and the variation (in effect size and timing) across neurons shown in the figures is actually a variation across neurons in different models rather than a variation of neurons within the same model. In networks with many units and reciprocal connections, the network dynamics might actually work against variation across neurons. The authors should discuss this. It would be great if it would be possible to show the same range of differences between neurons within a same model, but I will not insist on such a demonstration given that making such a larger model might require a substantial investment of time.*

We agree with the reviewer that in a model with many neurons and many incoming connections per neuron, dynamics across neurons may be similar due to “averaging” across all the inputs converging onto each neuron. However, in the limiting case where the neural population is large in size, but the number of active incoming connections per neuron is relatively small, substantial variation in inputs and in the response dynamics of individual neurons could persist. A detailed study of the precise conditions of connectivity and active network size needed to account for the observed data would be an interesting future direction for validating the model presented in this manuscript. We now mention this point in Discussion.

*5) "We cannot rule out the possibility that IT responses also contribute to V4 responses during the second transient peak. However, our IT recordings suggest this is unlikely because, as in V4, shape selectivity in IT is stronger for unoccluded than occluded stimuli". Is it conceivable that some IT neurons also have two phases in their response where the second phase is more pronounced in the presence of occlusion? It would be great if the authors could look for this possibility in the previous data set by Namima and Pasupathy, 2016? If the two phases are there it would strengthen the paper, but it would also be interesting if that is not the case.*

Our preliminary analyses reveal that only a few IT neurons (8/102) exhibit a second transient response peak that is generally smaller in amplitude than what we have observed in V4. Pending further analyses of the IT dataset, our current thinking is that the second transient response peak is much more prevalent and prominent in V4 than in IT cortex.

*6) Equations 4/5: I failed to see the logic of these equations, would it be possible to clarify this? Equation 9: what is _thr2_?*

We have revised the Methods section to clarify the logic behind Equations 4 and 5, and the termrthr2.

Equations 4 and 5 were designed to simulate the input to V4 units (e.g. Figure 10) with an onset latency of 30 ms, a strong initial transient response, and a gradually declining sustained response, collectively lasting ~500 ms. Specifically, a difference of Gaussian filter (k) was convolved with the ramp (R), then cubed, normalized and half-wave rectified (Equation 4) to produce the desired input. The ramp function (R), defined separately for the preferred (i = 1) and nonpreferred (i = 2) stimuli (Equation 5), increases monotonically with the% visible area (c) and declines over time with a support of 500 ms (30 < t < 530ms). Note that the precise function defininguFFi is not critical as long as it produces strong input signals for the preferred shape that decrease with increasing occlusion level, thus capturing the observed V4 neuronal response properties.

In Equation 9, rthr2 represents a threshold value on vlPFC activity. When vlPFC activity exceeds rthr2 (10 spk/sec, see Table 1), the steady state feedforward connection from V4 to vlPFC, w∞,ff, goes to 0, and any subsequent input from V4 will fail to activate vlPFC. This synaptic adaptation term was introduced to prevent the second response peak of V4 units from inducing a second response peak in vlPFC units (see Figure 10—figure supplement 1).

*7) I found Figure 8—figure supplement 6 confusing: how do you compute y/z for neurons with one peak?*

We now provide additional clarification in the legend of this figure supplement.

V4 neurons were classified as having one response peak if: i) their responses did not include a candidate second peak with a sizable amplitude and trough-to-peak modulation, or, ii) if their responses did include a candidate second peak that did not pass the statistical criterion. For 43 V4 neurons without a candidate second peak, we cannot compute y/z and these neurons are assigned y/z = 0 in Figure 8—figure supplement 6. For 12 neurons with a candidate second peak that did not pass the statistical criterion we can compute y/z and the values are as shown.